# Kinetic analysis of multistep USP7 mechanism shows critical role for target protein in activity

Robbert Q. Kim[1], Paul P. Geurink[2,5], Monique P.C. Mulder [2,5], Alexander Fish[1], Reggy Ekkebus[2,5], Farid El Oualid[3], Willem J. van Dijk[1], Duco van Dalen[2,6], Huib Ovaa[2,5], Hugo van Ingen[4,7] & Titia K. Sixma[1]

USP7 is a highly abundant deubiquitinating enzyme (DUB), involved in cellular processes including DNA damage response and apoptosis. USP7 has an unusual catalytic mechanism, where the low intrinsic activity of the catalytic domain (CD) increases when the C-terminal Ubl domains (Ubl45) fold onto the CD, allowing binding of the activating C-terminal tail near the catalytic site. Here we delineate how the target protein promotes the activation of USP7. Using NMR analysis and biochemistry we describe the order of activation steps, showing that ubiquitin binding is an instrumental step in USP7 activation. Using chemically synthesised p53-peptides we also demonstrate how the correct ubiquitinated substrate increases catalytic activity. We then used transient reaction kinetic modelling to define how the USP7 multistep mechanism is driven by target recognition. Our data show how this pleiotropic DUB can gain specificity for its cellular targets.

[1] Division of Biochemistry and Oncode Institute, Netherlands Cancer Institute, Plesmanlaan 121, 1066 CX Amsterdam, the Netherlands. [2] Division of Cell Biology II, Netherlands Cancer Institute, Plesmanlaan 121, 1066 CX Amsterdam, the Netherlands. [3] UbiQ Bio BV, Science Park 408, 1098 XH Amsterdam, the Netherlands. [4] Macromolecular Biochemistry, Leiden Institute of Chemistry, Leiden University, Leiden, the Netherlands. [5] Present address: Oncode Institute and Department of Cell and Chemical Biology, Leiden University Medical Center, Leiden, the Netherlands. [6] Present address: Tumor Immunology department, Radboud Institute for Molecular Sciences, Nijmegen, the Netherlands. [7] Present address: Bijvoet center, Utrecht University, Utrecht, the Netherlands. Correspondence and requests for materials should be addressed to H.O. (email: h.ovaa@lumc.nl) or to H.V.I. (email: H.vanIngen@uu.nl) or to T.K.S. (email: t.sixma@nki.nl)

Ubiquitination is an important post-translational modification (PTM) that influences protein fate in every cellular process[1,2]. This modification conjugates the C-terminus of ubiquitin (Ub) to a lysine residue on a target protein via an E1-E2-E3 cascade[3]. As Ub has 7 lysines and an available amino terminus it can be ubiquitinated itself, resulting in polyubiquitination through 8 different possible linkages[4]. These different ubiquitin marks generate distinct signals that determine the fate of the target protein, ranging from proteasomal degradation to cellular relocalisation or recruitment of complex partners[2,5,6].

Similarly to other PTMs, ubiquitination can be reversed, modulating and fine-tuning the ubiquitin signal[7]. Deubiquitination is carried out by deubiquitinating enzymes (DUBs) that hydrolyse the isopeptide bond between Ub and the target protein[8]. The activity of DUBs is tightly controlled[9] and their dysfunction can lead to serious diseases, such as cancer[10,11].

One of the most abundant DUBs is ubiquitin specific protease 7 (USP7, also known as HAUSP[12]). It has been implicated in several cellular processes ranging from DNA repair and apoptosis to suppression of regulatory T-cell function[13,14]. Mutations in USP7 have been shown to correlate with paediatric cancer[15,16] and the protein is actively targeted for cancer therapy[17–19], primarily for its nuclear functions, while USP7 haploinsufficiency leads to a neurodevelopmental disorder[20] through a cytosolic role.

USP7 is found in a variety of protein complexes, many of which contain an E3 ligase and its target[21]. In these complexes both the E3 ligase and its substrate are targets of USP7, like the substrate pair of E3 ligase MDM2 and target p53[22], the master regulator of the response to cellular stress[23]. This creates a situation where USP7 can either deubiquitinate and stabilise MDM2, promoting p53 ubiquitination and its proteasomal degradation[24,25], or target p53, preventing degradation and activating the apoptotic pathway[26]. The choice between these two targets is influenced by various other proteins shifting USP7 activity towards MDM2[27] or p53[28].

For the interaction with both MDM2 and p53, USP7 relies on its N-terminal TRAF (Fig. 1a) domain on USP7. This domain interacts with a TRAF recognition motif on the target proteins with a moderate affinity of ~10 μM[29,30], but does not affect the actual hydrolysis of the ubiquitin isopeptide bond on a minimal substrate[31]. The TRAF domain is connected to the adjacent catalytic domain (CD) through a flexible linker[30], allowing the CD to find and cleave off the ubiquitin from the target (Fig. 1a). This catalytic domain alone has low intrinsic deubiquitinating activity while full-length USP7 is a much more active DUB[32]. Crystal structures of this CD show that the *apo* state of the enzyme has an inactive conformation, with a misaligned catalytic triad[33]. When ubiquitin is bound, the catalytic triad (C223, H464 and D481) realigns into an active conformation, which involves significant changes in a loop above the active site. This 'switching loop' is essential for full activity of full-length USP7[31].

Located C-terminally of the CD are five ubiquitin-like (Ubl) domains which are essential for the increased activity of full-length USP7[31,32]. The three Ubl domains just downstream of the CD (Fig. 1a) do not influence the activity directly, but rather serve as a binding platform for interactors such as GMPS or DNMT1[31,34,35]. The last two Ubl domains with the activating tail (Ubl45), however, are indispensable for full activity of USP7: Ubl45 readily activates the CD as does the very C-terminal tail by itself, at high concentrations[36].

The Ubl domain region can adopt an extended conformation, as seen in the crystal structure of Ubl12345 (PDB: 2YLM), but has considerable flexibility, as shown by small-angle X-ray scattering (SAXS)[31]. Detailed biochemistry combined with SAXS analysis

led to a proposed mechanism where the Ubl domains curve and the Ubl45 domain 'folds back' onto the CD. The C-terminal tail then interacts with the 'switching loop', stabilising a catalytically competent conformation of USP7. Various mutations in either the tail or the loop substantiated this model[31]. The role of the C-terminal tail was further defined in a crystal structure of ubiquitin-bound CD linked to Ubl45. This showed how the C-terminal tail binds the CD, stabilising the 'switching loop'[36] in the active conformation. Intriguingly, fusion of just the C-terminal peptide to CD can reconstitute much of the activation, but from the structure it was unclear whether it was bound *in cis* or *trans*. This ambiguity prompted us to further investigate the role of the Ubl45 domain in this interaction and its effect on USP7 activity.

Most molecular studies on DUB activity utilise minimal substrates, focussing on the role of the ubiquitin moiety, essentially the product of the reaction. In the last years the focus has therefore shifted towards Ub-chains[4], uncovering chain-specificity of DUBs, which allowed relating them to distinct biological processes[37]. For USP7 the active conformation of the catalytic domain has only been observed in the ubiquitin-conjugated complex[33]. Ubiquitin alone is not sufficient to induce the rearrangement and a fusion at its C-terminus (such as ubiquitin aldehyde or a ubiquitinated substrate) is required for proper active site rearrangement[38].

The roles of target proteins, however have received relatively little attention in biochemical DUB analyses. Quantifying contributions of a realistic substrate requires a homogeneous, well-defined target. For p53, the interaction with USP7 has been described in detail[29,39], allowing generation of synthetic mimics of the substrate. Using such ubiquitinated p53 mimics as model targets, we investigate the effect of this more realistic substrate on USP7 activity in an in vitro setting.

With these chemical tools we address how a p53 model substrate interaction may modulate the activation process. Structural analysis suggests that monomeric USP7 undergoes an activation process that can be further improved by binding to a valid substrate. Using the p53 model substrate and global modelling of the experimental data, we could determine the order of events and quantify the steps involved in the USP7 ubiquitin hydrolysis cycle.

## Results

**USP7 activation requires the C-terminal tail in cis.** The activation of USP7 requires interaction between the CD and the C-terminal peptide (see Fig. 1a for domain definitions and nomenclature). The details of this interaction were described in a recent crystal structure (PDB: 5JTV) of Ubl45 and CD[36]. This structure clarifies how the activating C-terminal peptide binds, but the connection to Ubl45 was disordered, making it difficult to establish whether the C-terminus of USP7 binds into the activation cleft *in cis* or *in trans*.

When we analysed the ability of USP7 to form dimers in size-exclusion chromatography with multi-angle laser light scattering detection (SEC-MALLS), we observed no dimerization for full-length (FL) USP7 (injected at 20 μM, peak elutes at ~4 μM) and only partially for the construct used in the crystallization experiment (injected at 45 μM, monomer peak elutes at ~7 μM) (Fig. 1b). This suggests that at the much lower concentration found in cells (~0.3 μM[40]), USP7 is more likely to exist as a monomer.

We then checked what concentrations of Ubl45 are needed for *in trans* activation of CD (Fig. 1c). We find that *in trans* activation is possible, but only occurs at high concentrations, with an apparent $K_D$ of 110 μM (Fig. 1d). Consistently, a construct lacking the C-terminal tail (CD12345$^{\Delta C}$) can be activated by a

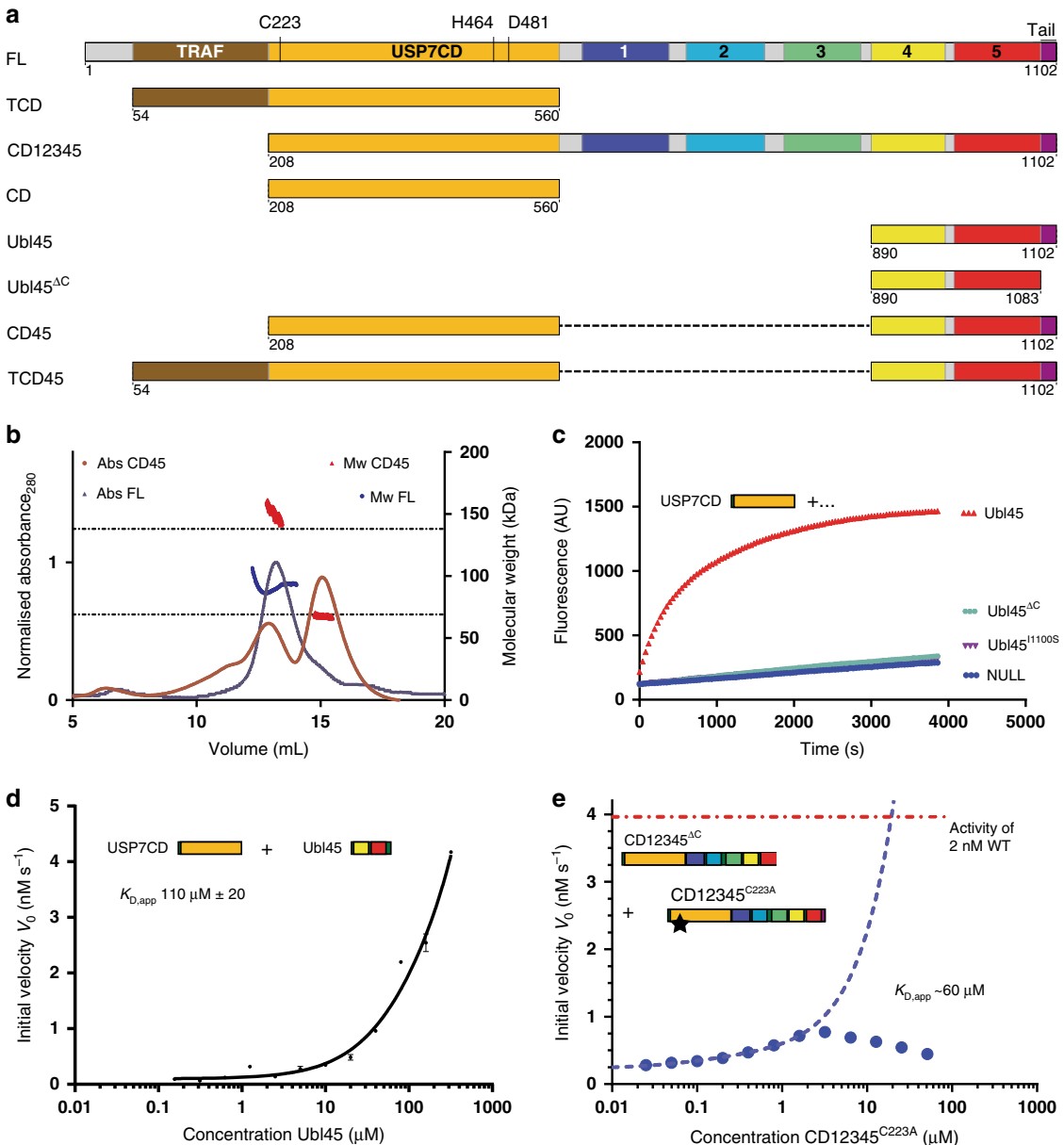

**Fig. 1** USP7 activation by the C-terminal tail happens *in cis*. **a** Schematic domain representation of USP7 and constructs used in this study. Active site residues and domain names are indicated: TRAF TRAF domain, CD Catalytic domain, 1–5 Ubl domains 1 through 5, tail. The activating C-terminal peptide (res. 1083–1102), marked in purple. The graphical representation of the constructs is used in other figures. **b** Analysis of USP7 constructs on SEC-MALLS shows monomer/dimer equilibrium. CD45 (100 μL of 45 μM) or FL (80 μL of 20 μM) were loaded on a Superdex 200 gel filtration column. Absorbance at 280 nm (dark red: CD45; dark blue: FL) was monitored and eluted peaks were analysed for molecular weight (red: CD45; blue: FL) by in-line MALLS. For CD45 the molecular weights of the monomer (69 kDa) and dimer (138 kDa) are indicated with the dotted line. **c** Activation of USP7CD by Ubl45 requires the very C-terminal tail. CD (20 nM) was mixed with Ubl45 variants as indicated and tested for DUB activity using UbRho. **d** USP7CD (20 nM) was incubated with a titration range of Ubl45, and tested in a deubiquitination assay as in 1c. These initial velocities were plotted against the concentration to yield an apparent $K_D$. Each data point is the mean ± SD of $n = 2$ measurements. **e** CD12345$^{\Delta C}$ (20 nM) has similar activity to CD and can be activated by the tail: when titrating in CD12345$^{C223A}$, which has no activity, the tail will activate the tailless construct upon dimerization. The activity readout shows that this dimerization-dependent activation of USP7 happens at micromolar concentrations in line with affinity of 1d. Binding of ubiquitin substrate to CD12345$^{C223A}$ causes an inhibitory effect above 2 μM, therefore only lower concentrations were used to extrapolate a $K_{D,app}$ (blue dashes). The red dotted lines indicate the activity of WT CD12345 for comparison

catalytically dead FL USP7, (CD12345$^{C223A}$) only at high concentrations, with an apparent $K_D$ of 60 μM (Fig. 1e). Both these apparent *in trans* activation constants are orders of magnitude higher than the concentrations (1–20 nM) that are sufficient for USP7 activity assays of full-length or CD-Ubl45 constructs[31]. We therefore conclude that, although *trans*

activation of USP7 is possible at high concentrations, it cannot be the predominant mechanism of its self-activation.

**Definition of the interaction interface between CD and Ubl45.** If USP7 acts as a monomer, this means that the interaction of CD and Ubl45 occurs *in cis*. The interaction of the C-terminal peptide

with CD, as determined by Rougé et al. in the crystal structure[36], is indisputable, but the positioning of the Ubl45 core did not seem as well-defined. We therefore wanted to investigate this interaction in solution. We used NMR spectroscopy to map the binding interface between CD and Ubl45 (Fig. 2, Supplementary methods). First we assigned the backbone resonances of Ubl45 (Supplementary Figure 1a) and performed prediction of secondary structure in solution. Since this matched the crystal structure (PDB: 5JTV, Supplementary Figure 1b), we concluded that the NMR conditions allow functional interaction analysis.

We then analysed the interaction of Ubl45 with CD. We titrated the unlabelled CD (42 kDa) into isotope-labelled Ubl45 (25 kDa) up to a ratio of 1:10 and followed changes in peak position or intensity (Fig. 2a, b). Upon addition of CD, we observe only very minor chemical shift perturbations (CSPs) (Fig. 2b), but a marked decrease in intensity for residues in the core of the Ubl45 domains (Fig. 2c). The size of the decrease agrees well with the formation of a 67 kDa complex with affinity as determined by SPR (see below and Supplementary Figure 2a). Under these conditions, residues with large changes in their chemical environment upon binding are expected to show an even more dramatic intensity loss (Supplementary Figure 2b-c). From the absence of such effect for residues that are predicted to be in the interface based on the crystal structure (Fig. 2d) or for any other site on the core surface, we conclude that the Ubl45 core is not involved in a single, specific interaction with CD. Only a few, very minimal CSPs are observed, localized to C-terminal tail residues, with the strongest shift seen for Y1093 (Fig. 2b). The tail residues however remain sharp and intense peaks throughout the titration, indicating that the tail is not immobilized on the CD surface.

This is surprising, since the tail is immobilised in the crystal structure, but this was solved in the presence of ubiquitin. Therefore, we wondered if ubiquitin could promote the binding of the tail. We generated a covalent complex with ubiquitin, as ubiquitin monomers bind poorly to USP7CD[38], We used a suicide probe, ubiquitin-propargyl (Ub-PA[41]) to generate CDUb (Supplementary Figure 3a). In an NMR titration of CDUb into labelled Ubl45, we observed interaction only through a decrease in peak intensity, which was nearly directly proportional to the equivalents of CDUb added. At 30% of CDUb added, peak intensity for all Ubl45 residues, including the C-terminal residues, was reduced by ~30% (Fig. 2e). This indicates that Ubl45 forms a tight complex with CDUb. Now also the tail, and particularly the C-terminal residues are immobilized. Meanwhile we do not observe significant CSPs. This absence indicates that the free protein is in slow exchange with the complex (75 kDa) (see also Supplementary Figure 2b-c).

Together, these results suggest that while the Ubl45 binds to CD, it does so in multiple, weak binding modes predominantly involving the Ubl45 core to form a dynamic complex. Our data further suggest that the presence of ubiquitin or a ubiquitinated target can induce the specific binding mode of the C-terminal tail as observed in the crystal structure.

**Interaction between Ubl45 and CD does not require the tail**. For full activity the C-terminal tail is essential, but its affinity for the catalytic domain could only be measured indirectly. In surface plasmon resonance (SPR) experiments, interaction was not detectable[31], but in an activation assay, the apparent $K_D$-value was estimated at ~1000 µM[36]. This is one order of magnitude weaker than the apparent $K_D$ determined for Ubl45 (110 µM, Fig. 1d). Our NMR results also suggested that the Ubl domains contribute to the binding and activation of the catalytic domain. To investigate the interaction between Ubl45$^{\Delta C}$ and the catalytic

domain we immobilised GST-USP7CD on the SPR chip, flowing over the tailless construct Ubl45$^{\Delta C}$. We were able to detect binding at high concentrations. Extrapolation of the curve, suggests a $K_D$ of 420 µM (Fig. 3a), similar to that observed for the C-terminal peptide interaction in the activity assay. This suggests that both Ubl45$^{\Delta C}$ and the tail bind weakly to the CD.

We then tested the affinity of CD for Ubl45 including the tail, and get an approximate $K_D$ of 280 µM (Fig. 3a), comparable to the tailless construct. This suggests that the C-terminal tail is not the main driving force for the interaction between the CD and Ubl45 as the affinity is similar with or without the tail. This result is in agreement with the lack of binding between the C-terminal peptide and CD observed in the NMR experiment (Fig. 2c) and earlier data[31], but seems at odds with the activating role of the tail in the activity assay. The NMR experiment seemed to suggest that ubiquitin needs to be present for immobilization of the tail on CD.

**Interaction between C-terminal tail and CD requires Ub**. To quantify the effect of ubiquitin on the binding of the activating C-terminal tail, we again used ubiquitin covalently coupled to the catalytic cysteine of the CD (CDUb). As CD on its own has low activity, the reaction between Ub-PA and CD had to be driven to completion using the *trans* activation of Ubl45. After incubation however, Ubl45 could not be separated from CDUb on gel filtration (Supplementary Figure 3a) requiring additional ion exchange chromatography. This indicates that complex formation between CDUb and Ubl45 is tighter than predicted, as a complex with a $K_D$ of 280 µM (Fig. 3a) generally dissociates during this type of gel filtration experiment.

We quantified the interaction by SPR flowing CDUb over GST-immobilised Ubl45 (Fig. 3b). Data analysis in EvilFit[42] identified a $K_D$ of 0.59 µM with a standard deviation of ±0.03 and a $k_{off}$ of 0.8 s$^{-1}$ (Supplementary Figure 3b) for the interaction between CDUb and Ubl45. The presence of ubiquitin in CD therefore increased the affinity 400-fold compared to CD only (Fig. 3c). As the C-terminal tail is necessary for activation, we hypothesised that it would directly facilitate the interaction with the intermediate, ubiquitin-bound, state. To test this, we immobilised the tailless construct (Ubl45$^{\Delta C}$) in our SPR experiment setup and flowed over CDUb with concentrations up to 80 µM (Fig. 3b). We could not detect any binding of Ubl45$^{\Delta C}$ in this experiment, suggesting that the Ubl domains can no longer bind ubiquitin-bound catalytic domain, contrary to *apo* CD (Fig. 3a). This would mean that, after ubiquitin binding, the increased activity depends exclusively on the C-terminal tail.

In agreement to this, the immobilised tail peptide (residues 1083–1102) interacted with CDUb with a $K_D$ of 2.6 µM, but showed no binding to the CD alone (Fig. 3d and Supplementary Figure 3c). This is analogous to what we found for Ubl45 (Fig. 3b) and in line with our NMR experiments where the tail did not show clear binding to CD alone (Fig. 2c) but was immobilized by CDUb.

Our results are in line with previously published NMR data that showed that a linked Ub is necessary to induce rearrangement of the catalytic site[38]. The binding of ubiquitin to CD apparently facilitates binding of the C-terminal tail (Fig. 3d). Together, these data explain how CD can still be activated by the C-terminal tail on its own, albeit with a lower resulting activity than the FL construct[36]. We conclude that once the ubiquitin-bound intermediate state is achieved, the C-terminal tail is sufficient for self-activation.

**Ubl45$^{\Delta C}$ promotes ubiquitin binding**. Knowing that the C-terminal tail has high affinity for CD only after ubiquitin binding

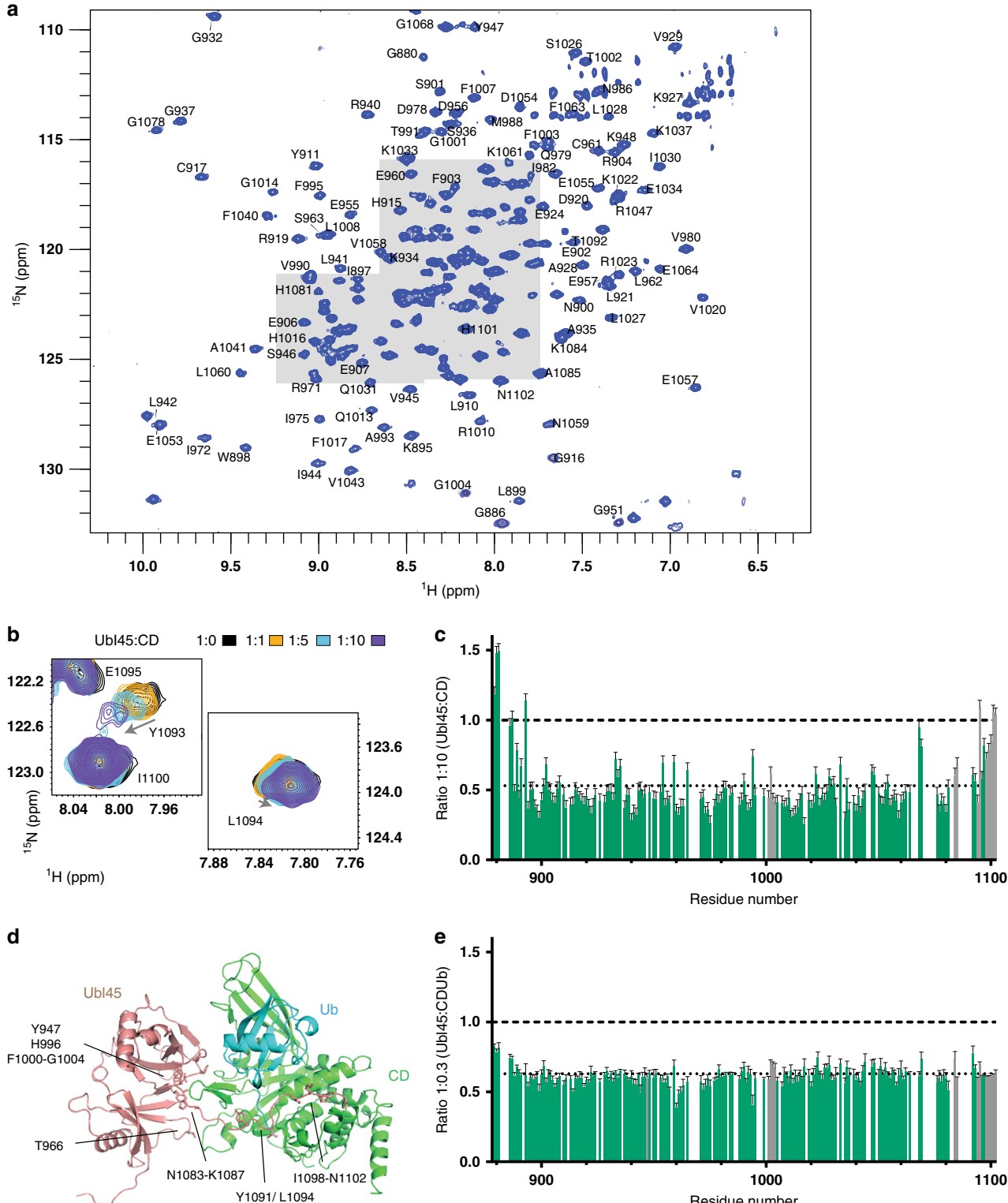

**Fig. 2** Interaction between Ubl45 and CD identified in solution using NMR. **a** The peak dispersion and resolution in the $^1$H–$^{15}$N correlation spectrum of Ubl45 (45 μM; 25 kDa; coloured blue) indicates a well-folded protein. Assignments are indicated, those for the crowded regions, indicated in grey, are shown in Supplementary Figure 1a. **b** The addition of 450 μM CD prompted very little chemical shift perturbations (CSPs). The biggest observed CSP of 0.019 ppm for Y1093 is illustrated in this zoom. **c** Upon titration of CD the peaks in the Ubl45 spectrum shows significant decrease in intensity. Here the intensity ratios between the *apo* spectrum (1:0) and the highest titration (1:10) are plotted against the residue numbers. The average is indicated by a dotted line, while residues that were found in the crystal structure to interact (**d**) are highlighted in the bar graph. **d** Structure and intermolecular interface in the Ubl45-CDUb structure (PDB: 5JTV[36]), showing contacts between both the tail and the core of Ubl45 to CD. Ubl45 residues that are within 4 Å of CD are shown as sticks and indicated. **e** The same intensity plot as in **c**, but now done for the titration with CDUb indicates that the tail now does get immobilised. For **c** and **e**, the bars are normalised against the overall intensity of all peaks of the *apo* spectrum

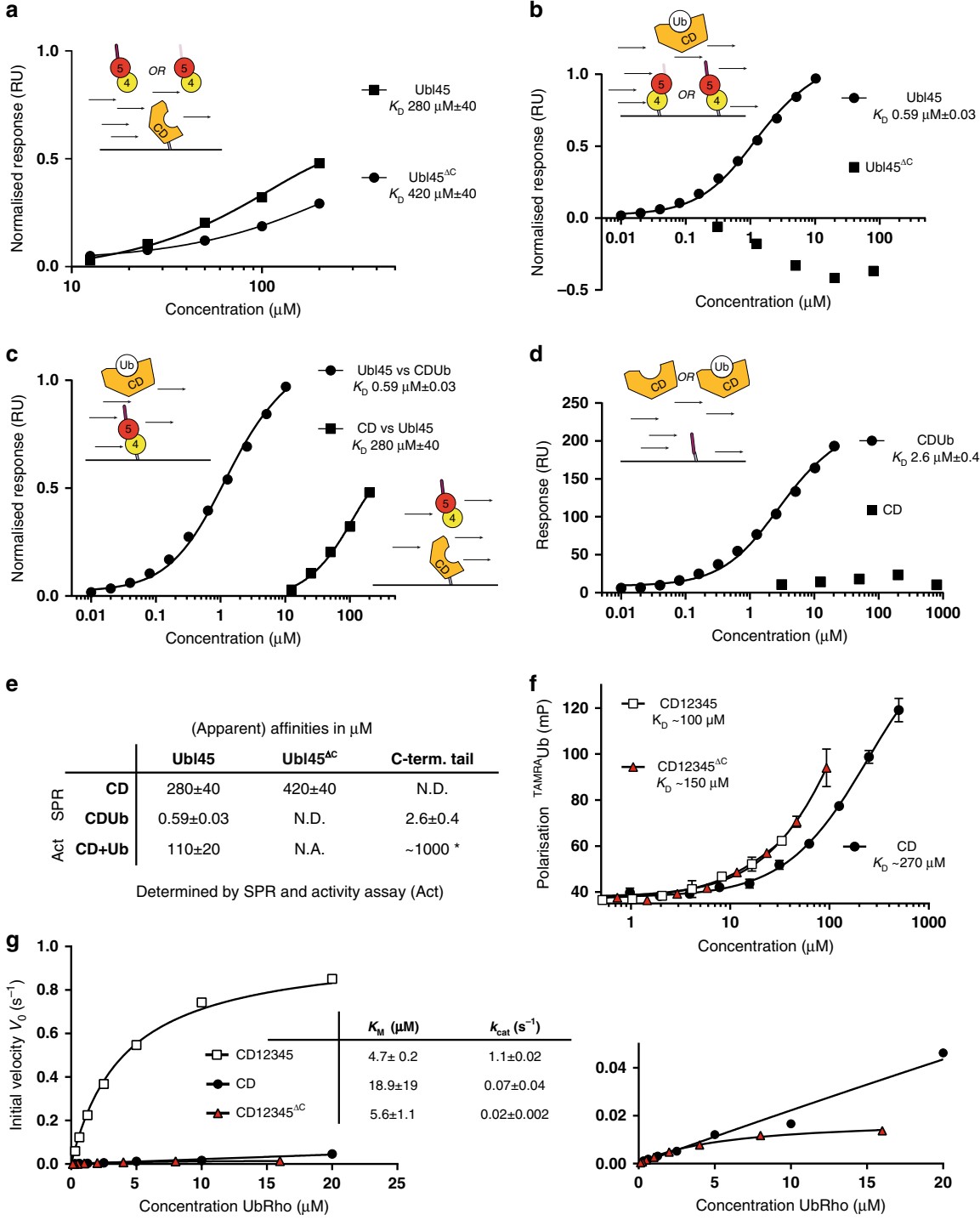

(Fig. 3e), we wondered if Ubl45$^{\Delta C}$ might affect the Ub binding. Such an outcome is consistent with our previous results indicating that in full-length USP7 self-activation increases the catalytic rate ($k_{cat}$) but also the $K_M$ (which is, on a minimal substrate, dominated by the affinity to ubiquitin), from >>35 µM to roughly 4 µM[31]. It may also explain why direct linkage of the C-terminal peptide to the CD almost, but not completely recapitulates the full-length activity[36].

We therefore tested ubiquitin binding qualitatively in a fluorescence polarization (FP) assay, following polarisation of TAMRA-labelled Ub upon incubation with various USP7

constructs (Fig. 3f). In this assay, we see the increased affinity for ubiquitin in the presence of the Ubl45 domain (when comparing CD12345 with CD only) even when the tail is absent (comparing CD12345$^{\Delta C}$ to CD). The data indicate that the presence of the C-terminal tail does not affect the CD affinity for Ub, in line with its lack of affinity for CD observed in the NMR (Fig. 2b) and SPR experiments (Fig. 3d).

We could confirm the Ubl45-induced increase in affinity of CD for Ub by analysis of the steady-state kinetics of these constructs in activity assays (Fig. 3g). When we fit Michaelis-Menten curves for CD, CD12345 and CD12345$^{\Delta C}$ we could see that presence of

**Fig. 3** Affinity of CD for Ubl45 increases with Ub present and is dependent on C-terminal tail. **a** SPR binding results indicate a weak affinity of USP7CD for either Ubl45 or Ubl45$^{\Delta C}$. CD was immobilized through GST on the chip and tested for binding with Ubl45 or Ubl45$^{\Delta C}$. Equilibrium binding values were plotted against concentration and fitted to get an estimated $K_D$. Responses were normalised using $B_{max}$ and standard deviation for resulting values is given. **b** The increased binding between Ubl45 and CDUb depends on the C-terminal tail. Ubl45 or Ubl45$^{\Delta C}$ was immobilized on the chip and the covalent CDUb complex was flown over. A fit was made using the equilibrium binding values yielding a $K_D$ of 590 nM for Ubl45, whereas no binding could be observed for Ubl45$^{\Delta C}$. Normalisation was carried out using $B_{max}$. **c** Comparison of the affinity of Ubl45 for CD or CDUb shows a remarkable increase. Curves from A and B are replotted to exemplify the change in $K_D$. **d** The C-terminal tail was immobilized using biotin and CD or CDUb was flown over to confirm that the tail interacts with the transition state (CDUb) only and not the *apo* CD. **e** Overview of affinities between Ubl45 and CD show that Ub enhances the binding of the tail. The values for the upper two rows are determined using SPR, see **a**–**d**. The values in the last row have been derived from activity assays, see Fig. 1d and[36] for the estimated affinity of the C-terminal tail (*). N.A. not applicable; N.D. No binding detected. **f** The presence of the Ubl45 domain is essential for increased affinity of CD for ubiquitin, but not the C-terminal tail. The affinity for ubiquitin was measured in an FP assay where TAMRA-labelled ubiquitin was incubated with various USP7 constructs. **g** Steady-state kinetics analysis of USP7 constructs indicates that the C-terminal tail mainly affects the catalytic rate, while the presence of Ubl45 without the tail enhances the affinity for the ubiquitin substrate. The Michaelis-Menten constant ($K_M$) and $k_{cat}$ were obtained by fitting the initial velocity data for various concentrations of UbRho. For **a**–**e**, obtained values are displayed ± SD after fitting. For **f**–**g** data points are the mean ± SD of $n = 2$ measurements

the C-terminal tail dramatically increases $k_{cat}$[31]; whereas, the Ubl45$^{\Delta C}$ is responsible for the increase in $K_M$ (compare CD and CD12345$^{\Delta C}$ in Fig. 3g).

**A multi-step mechanism for USP7 activity**. These data suggest that USP7 is likely to follow a multi-step mechanism during its catalytic cycle. In the first step, binding of ubiquitin is facilitated by the core of Ubl45, which does not involve the C-terminal tail. After binding of ubiquitin, conformational changes align the catalytic triad[33]. In this state, the affinity of the CD for Ubl45 is decreased (Fig. 3e), but the affinity for the tail is dramatically increased, allowing optimal orientation of the activating C-terminal tail to form the activated state. This mechanism is reminiscent of the classical-induced fit in enzymology, where binding of the substrate can activate the enzyme: here, ubiquitin-induced binding of the C-terminal peptide stabilizes the active CD conformation and promotes fast hydrolysis of substrate, which is observed as an increase in $k_{cat}$.

Next, we wanted to address the role of the target protein in the USP7 mechanism. To study whether interactions with a ubiquitinated target protein would affect USP7 activation we make use of a chemical biology approach.

**Role of the target p53**. We chose p53 as our model target protein (Fig. 4a), which has six lysines near the C-terminus that can be ubiquitinated[39] as well as motifs that can be recognized by the USP7 TRAF domain[29,43]. We made a synthetic toolbox of ubiquitinated C-terminal p53 peptides[44] and initial tests on these conjugates indicated that all six lysines could be cleaved by USP7.

We generated two versions of the ubiquitinated p53 peptides with K382 as the ubiquitination site, either with the TRAF recognition motif (p53Ub, res. 357–389) or without this region (p53$_{short}$Ub, res. 368–389, Fig. 4a). The ubiquitin attachment was varied to allow different assays: a suicide version with a vinylamide (VA) linkage (**I**)[45] that can bind covalently, like the better-known vinyl methyl ester[46,47], a non-hydrolysable triazole linkage (**II**)[48] and a cleavable native isopeptide linkage (**III**)[44,49], for both the short and long versions of the peptide (Fig. 4b).

Using the p53Ub$_{VA}$ suicide probes, we were able to assess the role of the TRAF interaction in substrate binding. Although both short and long versions of the probe reacted readily with CD12345 (Fig. 4c, Supplementary Figure 4), the full-length USP7, which contains the TRAF domain, showed increased complex formation, specifically for the p53Ub peptide that has the TRAF recognition sequence. This indicates that the TRAF recognition sequence promotes the USP7 interaction with the model

substrate. In the remainder of this report we will focus on the long peptide.

Next, we used the p53Ub$_{VA}$ probe to generate a non-hydrolysable complex with TCD, a construct that lacks all Ubl domains (Fig. 1a) to address whether the p53 interaction affects the interaction of CD with Ubl45. Using SPR, the complex was flowed over immobilised Ubl45 and we could determine the affinity between Ubl45 and TCD-p53Ub (Fig. 4d, Supplementary Figure 3d). With a $K_D$ of 2.9 µM it is similar to the affinity found for the C-terminal tail interacting with CDUb (2.6 µM, Fig. 3d). Apparently the presence of the TRAF-p53 interaction does not further change the interaction with Ubl45. This is in line with the fact that the presence of the TRAF domain does not affect activity on a minimal substrate[31] or the association of the activating Ubl45 domain.

**A second p53 interaction**. To further investigate the role of the p53 peptide interaction with the TRAF domain in the activation process, we decided to look at the affinity between USP7 and the model target. As previous reports have alluded to an additional binding site (other than TRAF) for the USP7 targets p53 and MDM2 in the C-terminal domains of USP7[50], we first assessed the binding of the peptide to USP7 constructs in a FP assay (Fig. 5a). These direct binding assays with $^{TAMRA}$p53 peptide confirmed the presence of an additional p53 binding site and map it to the Ubl45 domains, without requiring the C-terminal tail (compare FL$^{\Delta C}$ and TCD45 to TCD). Interestingly, this additional interaction site depends on the TRAF domain since CD12345 alone does not bind the peptide at these concentrations (Fig. 5a). These results suggest an extended binding interface between the TRAF-CD and the p53 peptide that is aided by Ubl45, but the hypothesis of a second, very weak, binding site within Ubl45 cannot be excluded.

To then assess a potential increase in affinity for the ubiquitinated substrate, we used the non-hydrolysable compound (Fig. 4b; p53Ub$_{inh}$) as inhibitor in an activity assay, on minimal substrate UbRho (Fig. 5b). We found that both the TRAF domain (compare TCD and CD) as well as Ubl12345 improve the IC$_{50}$ independently (Fig. 5c). However, the full-length construct displays a further avidity effect, resulting in an IC$_{50}$ of 16 nM, ~60-fold better than either TCD or CD12345. This underlines that the TRAF domain, CD and Ubl12345 all contribute to the effective substrate (p53Ub) recognition and that the sum of these interactions yields a tight, effective interaction.

**Visualisation of the multi-step enzymatic mechanism**. As both the ubiquitin acceptance (aided by Ubl45) and the target

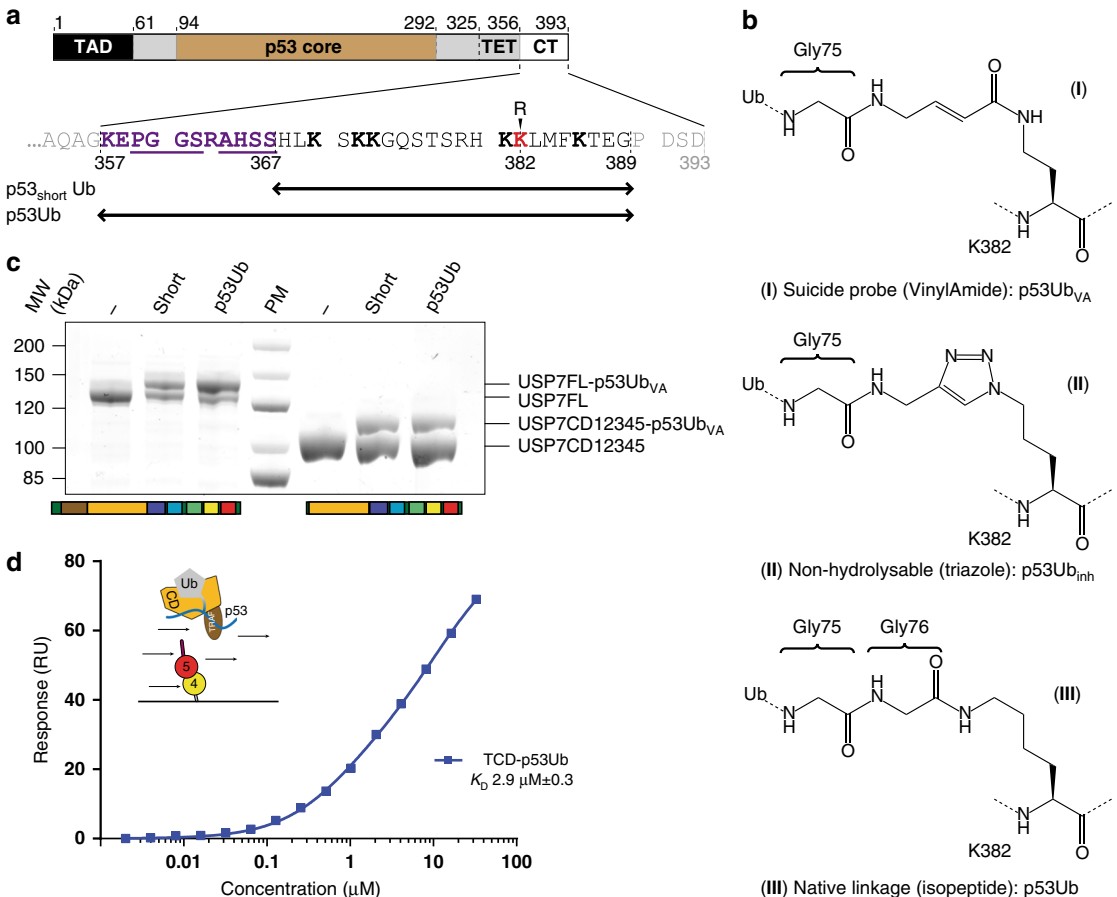

**Fig. 4** Development of synthetic p53-derived substrates for USP7. **a** Schematic domain structure of p53 with trans activation domain (TAD), core region, tetramerization domain (TET) and C-terminal region indicated (CT). Close-up of CT highlights lysines (bold) known to be ubiquitinated[39] and the TRAF recognition motif (underlined)[29,43]. K382 (red) is used as target lysine in the synthetic p53Ub reagents, underneath the sequence coverage of the peptides is indicated. **b** Three synthetic p53Ub-peptides are generated with different linkage types: covalently binding (vinylamide linkage, (**I**)), uncleavable (triazole linkage, (**II**)) and cleavable (native linkage, (**III**)). **c** The affinity of USP7 for the covalent-binding p53Ub$_{VA}$ increases if the TRAF domain is present. USP7 constructs with and without TRAF domain have been incubated with the probe for 15 min at RT and analysed on a Coomassie gel. **d** The affinity of Ubl45 for the substrate-bound TCD is similar to CDUb (2.9 μM compared to 0.59 μM, Fig. 3c). Around 50 units of Ubl45 were immobilized on the chip before a titration range of p53Ub-bound TCD was flown over. Their equilibrium binding values were plotted and fitted to get the displayed $K_D$-value, with the SD (±)

recognition positively influence deubiquitination, we wanted to explore how these collaborate during the deubiquitination process and whether there is a defined order of events. To this end, we utilised the synthetic ubiquitinated p53 target with a native linkage (Fig. 4b; p53Ub) and a fluorophore at the N-terminus of the p53 peptide to allow tracking of substrate and product[44]. We monitored the substrate during its hydrolysis in an FP assay where 100 nM of TAMRA-labelled reagent is incubated with various USP7 constructs (Fig. 6b for FL, Supplementary Figure 5-7b for TCD, CD12345 and CD). Both FL and CD12345 could readily hydrolyse the substrate, resulting in a drop of the FP signal. The other two constructs, CD and TCD, required higher concentrations in order to see a decrease in FP signal, while the TCD construct actually started out with an increased signal (Supplementary Figure 5b). This increased signal would be a result of binding, as the TRAF domain increases the affinity for the p53-substrate, but the rate of catalysis is still low for TCD. Although these experiments efficiently monitored substrate hydrolysis, we were interested in the early events that could not be caught in our plate reader setup.

To get insight into the very early phase of the reaction, we decided to use a stopped-flow setup (Fig. 6c, d). We followed the reaction by fluorescence polarization, which is sensitive to the size of the complex (as this affects the tumbling rate and thus polarisation), and by fluorescence intensity, which responds to changes in local conformation affecting the fluorophore. As these experiments were performed under near-equimolar amounts of enzyme and substrate, we also measured binding to the product TAMRAp53 peptide in this stopped-flow setup (Fig. 6e, f).

In the stopped-flow anisotropy data, we could detect an increase in signal when we titrated FL into TAMRAp53Ub (Fig. 6c). After this binding phase (0.02–0.2 s) we observe a decrease in anisotropy, indicating a second phase (0.2–2 s) indicative of hydrolysis. The signal however does not drop below baseline for the highest concentration, indicating retention of the product, in line with our p53 peptide binding data (Fig. 6e).

For the constructs CD and CD12345 we can hardly detect the TAMRAp53Ub binding phase (Supplementary Figure 6, 7) or the decrease in anisotropy, indicating that these require the presence of the TRAF domain. When TRAF is present, in TCD, both binding (increase in anisotropy) and hydrolysis (decrease in anisotropy) are visible (Supplementary Figure 5c), but the decrease only occurs after a lag phase (>5 s), indicating the presence of intermediate states between binding and hydrolysis.

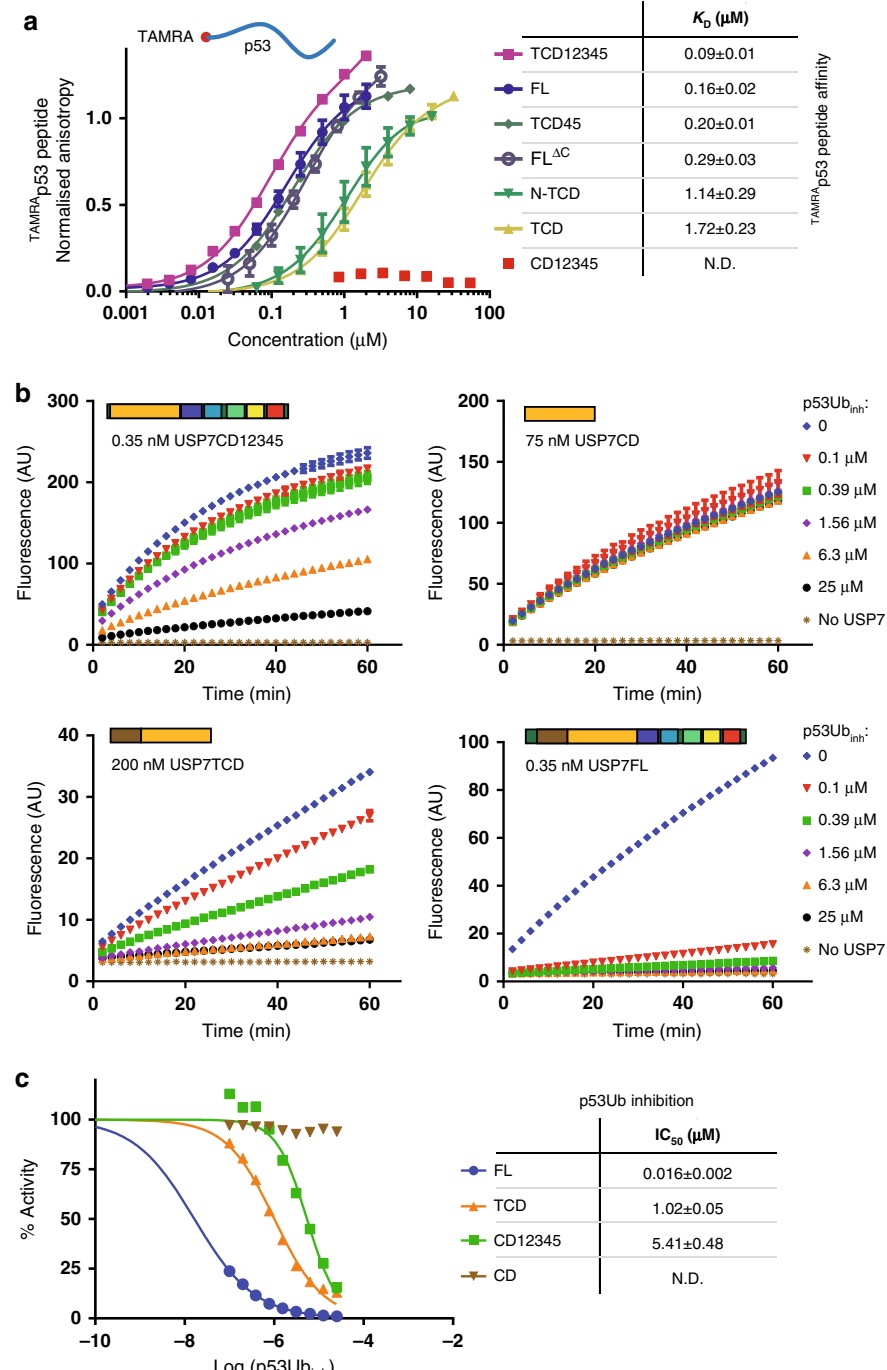

**Fig. 5** The p53-derived tools stress the importance of both the TRAF and Ubl45 domain. **a** Binding of the TAMRA-labelled p53 peptide is determined by incubating 25 nM of $^{TAMRA}$p53 with a dilution series of various USP7 constructs in FP assays. The obtained $K_D$ values for each USP7 construct are stated and indicate that in presence of the TRAF domain, the Ubl12345 domain affects recognition. **b** The non-hydrolysable p53Ub construct acts as an inhibitor for USP7 in deubiquitination assays. USP7 constructs, corrected for their activity (CD12345: 0.35 nM, TCD: 200 nM, CD: 75 nM, USP7FL: 0.35 nM) were incubated with increasing amounts of p53Ub$_{inh}$ and analysed for activity in a deubiquitination assay using a single concentration of UbRho. The raw data indicate that both the TRAF domain and the C-terminal Ubl domains increase the affinity towards the p53 construct. For clarity's sake a limited number of concentrations is shown. **c** The observed activity for USP7 constructs after incubation with p53Ub$_{inh}$ (see B), is plotted against the concentration of inhibitor used. Fitting yielded IC$_{50}$ values and standard deviation as shown. The data points for **a**, **b** are the mean ± SD of at least $n = 2$ experiments. All reported values include the SD (±)

This step is more explicitly visualised in the intensity data from these stopped-flow experiments. For TCD (Supplementary Figure 5), with the long delay between binding and hydrolysis, we see a significant decrease in intensity in this delay (Supplementary Figure 5d), which we interpret as a conformational change in the protein (see Supplementary methods). For full-length USP7 a minor intensity increase (Fig. 6d) occurs, with a slight delay (0.05–0.5 s) relative to the binding phase. This suggests that a further conformational change affects the intensity signal, which we interpret as binding

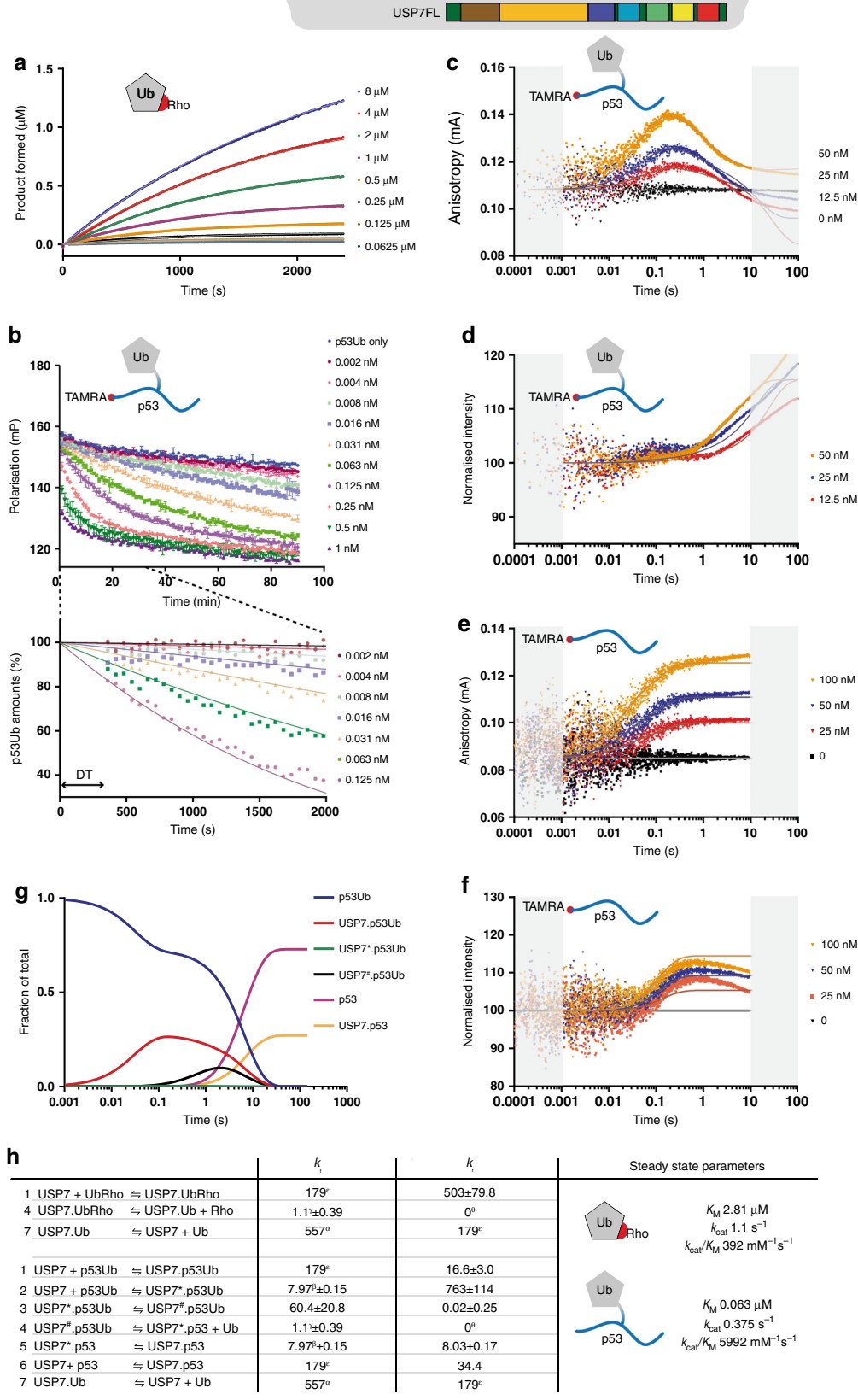

| | | $k_f$ | $k_r$ | Steady state parameters |
|---|---|---|---|---|
| 1 | USP7 + UbRho ⇌ USP7.UbRho | 179$^\varepsilon$ | 503±79.8 | $K_M$ 2.81 µM |
| 4 | USP7.UbRho ⇌ USP7.Ub + Rho | 1.1$^\gamma$±0.39 | 0$^\theta$ | $k_{cat}$ 1.1 s$^{-1}$ |
| 7 | USP7.Ub ⇌ USP7 + Ub | 557$^\alpha$ | 179$^\varepsilon$ | $k_{cat}/K_M$ 392 mM$^{-1}$s$^{-1}$ |
| 1 | USP7 + p53Ub ⇌ USP7.p53Ub | 179$^\varepsilon$ | 16.6±3.0 | $K_M$ 0.063 µM |
| 2 | USP7 + p53Ub ⇌ USP7*.p53Ub | 7.97$^\beta$±0.15 | 763±114 | $k_{cat}$ 0.375 s$^{-1}$ |
| 3 | USP7*.p53Ub ⇌ USP7$^\#$.p53Ub | 60.4±20.8 | 0.02±0.25 | $k_{cat}/K_M$ 5992 mM$^{-1}$s$^{-1}$ |
| 4 | USP7$^\#$.p53Ub ⇌ USP7*.p53 + Ub | 1.1$^\gamma$±0.39 | 0$^\theta$ | |
| 5 | USP7*.p53 ⇌ USP7.p53 | 7.97$^\beta$±0.15 | 8.03±0.17 | |
| 6 | USP7+ p53 ⇌ USP7.p53 | 179$^\varepsilon$ | 34.4 | |
| 7 | USP7.Ub ⇌ USP7 + Ub | 557$^\alpha$ | 179$^\varepsilon$ | |

of the C-terminal tail, read out by rearrangement of the TAMRA label. The non-synchronicity of the events in the anisotropy and intensity experiments suggests that multiple steps are involved in the hydrolysis mechanism.

**Kinetic analysis of USP7 activity on p53 model substrate**. To model these multiple steps we describe every phase, with as few reaction steps as possible. We imported the raw stopped-flow, FP and activity data into KinTek[51] and scaled the data based on the

**Fig. 6** A quantitative kinetic model for USP7 enzymatic activity through global fitting. Icons in each panel indicate the substrate used. The lines describe the fit of the data in the various experiments performed: **a** Minimal substrate activity assay of USP7FL (1 nM), using a dilution range of UbRho. **b** FP enzyme activity assay on $^{TAMRA}$p53Ub (100 nM). The amounts of USP7 are indicated. After conversion to p53Ub amounts (lower panel) a delay time (DT) was introduced. **c** Stopped flow FP enzyme activity assay on $^{TAMRA}$p53Ub (50 nM), the anisotropy signal allows observation of the early binding and hydrolysis phases. Areas marked in grey were not included in the fit as they represent the mixing time (<0.001 s) or a timescale where bleaching effects start to dominate (>10 s). **d** Intensity readings of the experiment in **c** indicates a change in chemical environment upon binding of substrate. **e** Like **c** for using peptide only (25 nM p53) shows equivalent binding phases as the full substrate. **f** Intensity readings of the experiment in **e**. All stopped-flow experiment are an addition of $n = 10$ separate measurements. **g** Behaviour of p53-substrate states during overall model in an equimolar (1–1; 50 nM) ratio of enzyme and substrate. Intermediate states as in **h**. **h** Model used for KinTek fitting with kinetic constants obtained. USP7FL indicated as USP7, intermediates with # or *. For binding steps (Step 1, 6 and 7) on-rates are in $\mu M^{-1} s^{-1}$ and off-rates in $s^{-1}$. Rates for conformational changes (Step 2, 3 and 5) are in $s^{-1}$. The forward reaction for enzymatic hydrolysis (Step 4) is in $s^{-1}$, but the reverse step (labelled with θ) was assumed irreversible (fixed at 0 $\mu M^{-1} s^{-1}$). Equation constants with matching Greek characters (α, β, γ and δ) were linked in the refinement. The on-rate for binding steps (labelled with ε) is diffusion-controlled, determined separately and fixed during modelling. These on- and off-rates reflect the optimal ratio that models the individual steps with their respective SD (±), as the experiment does not have sufficient resolution to fully resolve rates. For both UbRho and p53Ub the resulting steady-state parameters were calculated to allow for a direct comparison (last column)

negative controls (Supplementary methods). This rendered the data interpretable by KinTek modelling with a minimal set of reaction equations. To be able to model the non-synchronous changes in the FP and intensity signals we introduced intermediate steps in the reaction (Fig. 6h). These include binding events (Steps 1, 6 and 7), conformational changes (Steps 2, 3 and 5) and the enzymatic hydrolysis (Step 4). The introduction of these steps allowed a good fit to the data and made it possible to derive rate constants ($k_f$ and $k_r$) for every step of the mechanism (Fig. 6h).

For the shorter constructs, introduction of one intermediate step between the binding (Step 1) and the hydrolysis step (Step 4, Supplementary Figure 5, 6, 7) was sufficient to fit the experimental traces. For the full-length construct however a second intermediate step was necessary to match the model to the experimental data.

The order of release of the reaction products p53 and Ub could not be determined based on activity data alone. Therefore, we used their respective affinities (Fig. 3f, 5a) to set the order of release. This order, with later release of the p53 peptide, also allowed adding affinity data for the secondary binding site to the model, as induced by Ubl45 (Fig. 5a), which fitted well in the stopped-flow data (Fig. 6e, f).

The modelling of the experimental curves allowed us to tease apart the various steps that USP7 performs in catalysis and reveals how the different domains affect the target processing. For instance, the intensity decrease observed upon p53Ub binding by the TCD construct (Supplementary Figure 5d) is not seen for FL (Fig. 6d). We can interpret this absence as an effect of the 'folding back' of the Ubl domains towards the CD[31] and changing the p53 interaction[50]. Likewise, the long delay time, seen for TCD, is much shorter for constructs that still contain the Ubl domains. As these domains activate CD, it seems reasonable to assume the delay time in TCD is required to remodel the catalytic site into an active conformation[33] without help of the Ubl domains.

The KinTek analysis results in a model where we can quantify each component (Fig. 6g) and reaction kinetics (Fig. 6h). In the first step, anisotropy changes are interpreted as binding of substrate p53Ub (Step 1; Fig. 6h). This is followed by multiple changes in intensity which are interpreted as conformational changes (Steps 2 and 3). Next, the intensity rises and the anisotropy decreases (Step 4), interpreted as hydrolysis and release of the ubiquitin product. Further intensity changes (Step 5) take place before p53 peptide release returns USP7 to the ground state (Step 6).

**Validation and evaluation of the kinetic model.** Our kinetic model fully agrees with the order of events observed in NMR and

SPR analysis in Figs. 2–3. We decided to test whether this could be used quantitatively as an independent control. We applied the model to fit the analysis of the minimal substrate, UbRho (Fig. 6a, Supplementary Figure 5a, 6a, 7a). Besides validating the kinetic model, this would also allow better definition of the rate constants by co-refining the values within KinTek. The intermediate states were used in the fitting, making a direct comparison between the ubiquitinated target protein and the minimal substrate possible. For the FL construct the efficiency of the reaction precluded fitting the intermediate steps in the minimal substrate analysis, so we only used steps 1, 4 and 7 (Fig. 6h).

The validity of the fitted constants was then analysed using the FitSpace module of KinTek[52]. Here we found that only TCD and FL data had sufficient amplitudes to allow for a full statistical analysis (Supplementary Figure 8). To avoid overfitting within this analysis, we linked rate constants (Supplementary Figure 8a, d), testing the statistical relevance for their ratios rather than their absolute values. The overall result indicated a well-constrained model where Step 2, the catalytic rearrangement, is the rate-limiting step. Finally, we converted the rate constants into steady-state kinetics parameters using the appropriate formula for one (Supplementary Figure 8f) or two (Supplementary Figure 8c) intermediates[53] (Supplementary Figure 5g, 6f, Fig. 6h). This yielded $K_M$ and $k_{cat}$ values for all steps. The values for the minimal substrate are similar to those determined previously[31], validating the descriptions used in our modelling.

**USP7 activity is driven by target recognition.** The conversion to steady-state parameters allows for easy comparison of USP7 activity on minimal substrate and the p53 model substrate. The combined increased activity ($k_{cat}/K_M$) is ~11,000-fold, for FL on p53Ub relative to CD on UbRho. Interestingly, on the p53Ub substrate $k_{cat}$ is slightly diminished as soon as a TRAF domain is present, but this is offset by the improved target recognition (here expressed as $K_M$) leading to substantially increased processivity. These findings indicate that, although studies on a minimal substrate are essential in studying the enzymatic mechanism, using a realistic substrate can give better understanding of the working of a DUB and its possible regulation.

**Discussion**

Here we studied USP7 self-activation by its C-terminal peptide and its target protein. We show that although *trans* activation by self-association is possible at high concentrations, the normal USP7 self-activation happens *in cis*. We show that interaction of Ubl45 with the CD promotes ubiquitin binding and only this promotes the correct positioning of the C-terminal peptide next

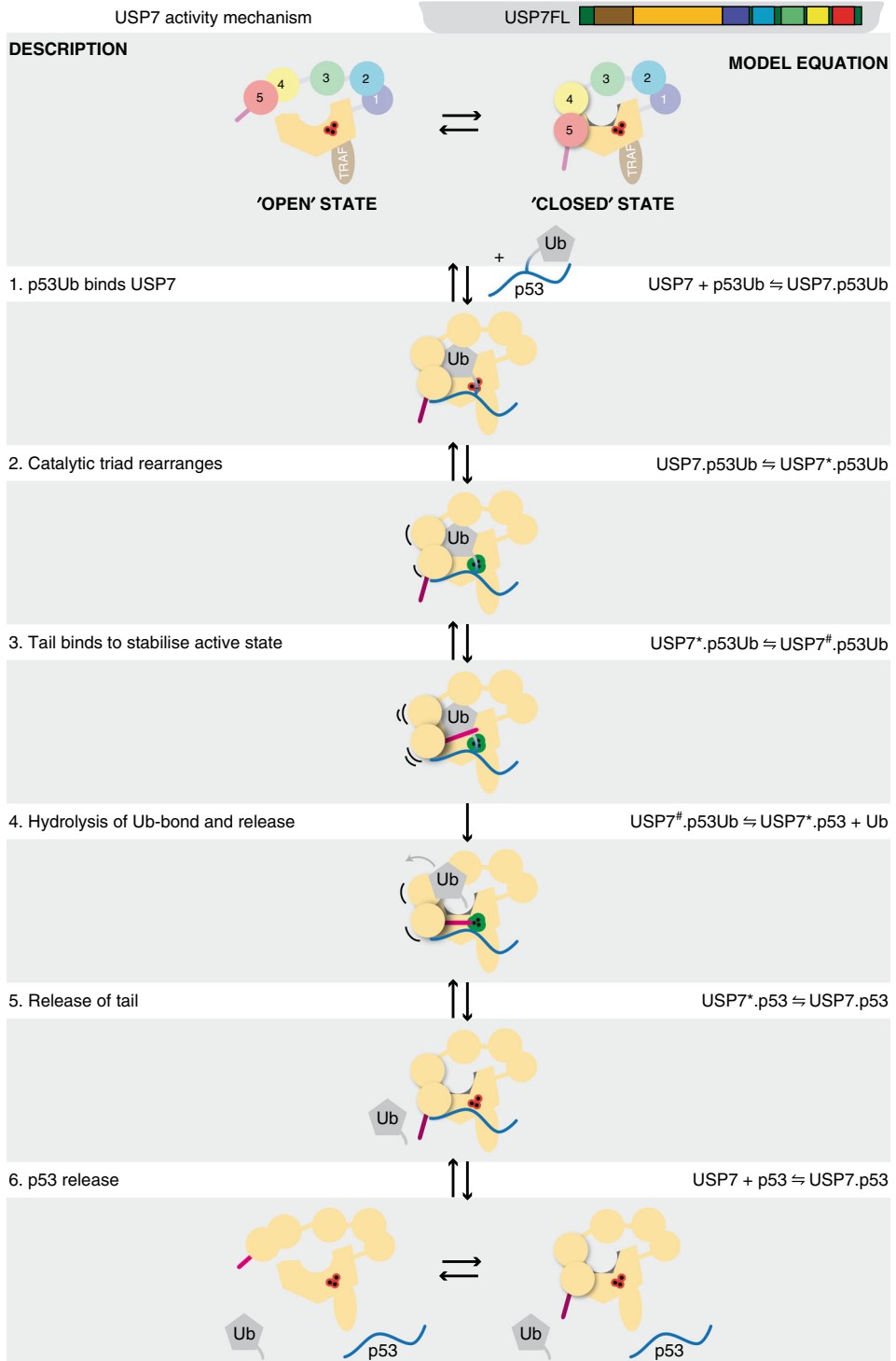

**Fig. 7** Kinetic model for USP7 mechanism on p53Ub The kinetic model (equations) and their interpretation are depicted schematically. The weak affinity between Ubl45 and CD suggests that free USP7 is in equilibrium between an 'open' and 'closed' state[31]. The p53Ub substrate is bound by TRAF and CD, as well as an additional binding site that depends on the Ubl domains (1). Ubiquitin binding induces a rearrangement of the catalytic triad[33] (2), which dramatically increases the affinity the activating C-terminal tail, but diminishes the contact between CD and Ubl45. Binding of the tail peptide (3) stabilises the active state. This promotes the hydrolysis of the isopeptide bond, allowing Ub release (4). This in turn diminishes affinity for the C-terminal tail, causing its release (5) and the subsequent release of the p53 peptide (6) to return USP7 to the ground state

to the catalytic site. Thus self-activation takes place in multiple distinct steps. Next we showed how the substrate protein strengthens activation and provided a kinetic model for the cooperative activation process.

The combination of our findings allows us to generate an updated model for the USP7 mode of action (Fig. 7). Ubiquitinated targets associate initially with the TRAF domain (Step 1, Fig. 7) and this binding is improved by the additional p53 binding

site, induced by presence of the Ubl domains (Fig. 5c). The target association brings the attached ubiquitin in close proximity to the catalytic domain, overcoming its poor affinity for ubiquitin, enhanced by the Ubl45 domain (Fig. 3f). The binding of ubiquitin into the active site (Step 2) not only induces rearrangements of the catalytic triad[33], but also reduces intramolecular interaction with Ubl45 (Fig. 3e) and promotes binding of the activating C-terminal tail, through a dramatic increase in affinity (Fig. 3) where it stabilises the active conformation (Step 3). In this activated state the hydrolysis of the isopeptide bond (Step 4) occurs much faster than for CD only[31].

After hydrolysis, release of products takes place, which we modelled according to their respective affinities. In full-length USP7 the leaving ubiquitin has a poor affinity compared to the p53 peptide (Fig. 5c), so we expect Ub to leave first (Step 4), leaving p53 bound to USP7, enabling a change due to the additional binding site (Step 5). The p53 release (Step 6) is modelled here as the last step in order to let USP7 return to the ground state, but given the tight interaction (160 nM) and the protein concentrations found in cells the p53-USP7 complex may last longer in vivo. Another ubiquitinated substrate or additional regulatory step could be required to perturb this complex and release p53.

With this model we assumed a sequential order of reactions (Supplementary methods) and we could not model all steps explicitly (Supplementary Figure 7). Nevertheless, the modelled intermediate steps agree very well with our SPR experiments that were not used for the model (Fig. 3). Based on the model we can separate intermediates in both time and place, allowing to connect species tested by SPR to states found in the kinetic model. Thus we see that Ubl45$^{\Delta C}$ is responsible for the increased $K_M$ whereas the C-terminal tail for the faster $k_{cat}$[31].

It is clear from our data that USP7 activation follows a multistep activation scheme that generates high specificity for the target. As USP7 interacts with many different targets[21], such a mechanism could make sure activity is targeted to the right substrate at the right time. Our results indicate that the substrate recognition collaborates with the intrinsic self-activation.

The complexity of the self-activation provides regulatory opportunities through external factors. One example is hyperactivation by GMPS[31], but other binding partners, such as ICP0 and DNMT1[35,54] and/or post-translational modifications may further affect activity.

Interestingly, the p53 peptide collaborates with the C-terminal domains through the additional binding site that we quantified. An earlier report suggested binding in the Ubl domains to both p53[50] and MDM2[36], but whether such a bipartite binding of substrates by USP7 is a common theme in other substrates remains to be investigated. Further definition of the different interactions would be needed to explain why USP7 usually prefers MDM2 over p53[25,55].

Our data provide opportunities for specific targeting in drug discovery programmes: both the secondary substrate binding site and the self-activation by Ubl45 are allosteric sites of interest. Working out the specifics of the interaction, using our NMR backbone assignment for Ubl45 and the recent assignment of CD[56], can be helpful in this process. Better molecular understanding of this interaction would help to design inhibitors specifically targeted at USP7 self-activation.

In this study, we employed both a model substrate and the minimal substrate to assess the USP7 mechanism of action. The usefulness of the ubiquitinated model substrate is not only illustrated by our findings on the activity effect of the TRAF domain, it also allowed us to monitor intermediate steps of the reaction and the order in which they occur. Combining these chemical ubiquitin tools with a domain-by-domain approach we could

pinpoint what part of USP7 is important in which part of the hydrolysis cycle. The results highlight the importance of the target protein and hopefully these insights will allow for the development of more specific USP7 inhibitors, targeting USP7 activity on specific substrates.

## Methods

**Constructs and mutations.** USP7 constructs (Fig. 1a) were based on the codon-optimized sequence (Addgene, #63573)[31]. USP7-TCD was cloned into pGEXNKI-GST-3C using ligase-independent cloning[57]. Constructs lacking the C-terminal tail were made by introducing a stop codon at residue 1083 using site-directed mutagenesis. Mutation constructs were introduced using partially overlapping primers and Phusion Flash polymerase (Thermo Fisher). All clones were sequence-verified and agreed with the published sequence[12].

**Protein expression.** USP7 constructs that included the TRAF region were expressed in *Escherichia coli* BL21 Rosetta2 (DE3) using Terrific Broth medium and overnight induction using 0.2 mM IPTG at 18 °C. Other USP7 constructs were expressed in *E. coli* BL21 cells using overnight auto-induction in LB at 18 °C[58].

Isotope-labelled USP7-Ubl45 intended for interaction analysis by NMR was expressed in *E. coli* BL21 cells using M9 minimal medium supplemented with $^{15}NH_4Cl$ (CortecNet), glucose, vitamin mix and micronutrient mix[59]. For three-dimensional NMR experiments $^{13}C$-glucose (CortecNet) was used. To acquire deuterated sample, $D_2O$ (CortecNet) was used to make the medium. Cells were grown in 5 mL LB from a single colony and transferred to a 50 mL minimal medium preculture after washing to grow overnight at 37 °C. The preculture was dispensed in 4 L minimal medium and cells were grown until $OD_{600}$ reached 0.6. Protein expression was then induced overnight at 18 °C by addition of 0.2 mM IPTG.

**Protein purification.** Expressed proteins were isolated from the lysate using Glutathione Sepharose 4B beads (GE Healthcare) in GST buffer (50 mM HEPES, pH 7.5, 250 mM NaCl, 1 mM EDTA, 1 mM DTT). After elution, using GST buffer with 15 mM GSH added, the GST tag was removed using 3C protease under dialysis against PorosXQ buffer A (20 mM HEPES, pH 7.5, 50 mM NaCl, 1 mM DTT). The sample was subsequently applied to PorosXQ anion exchange (Thermo Fisher) and eluted using a gradient of buffer B (20 mM HEPES, pH 7.5, 1 M NaCl, 1 mM DTT). After analysis appropriate fractions were pooled, concentrated and further purified on a Superdex gel filtration column (GE Healthcare) using GF buffer (20 mM HEPES, pH 7.5, 100 mM NaCl, 1 mM DTT). The peak fractions were pooled, concentrated up to 10 mg mL$^{-1}$ and flash frozen[34].

**MALLS experiments.** Purified protein was run on a Superdex 200 gel filtration column (GE Healthcare) using GF buffer (20 mM HEPES, pH 7.5, 100 mM NaCl, 1 mM DTT) in line with a MiniDawn Tristar (Wyatt Technologies) Multi-Angle Laser Light Scattering (MALLS) detector, connected to a Shodex RI 101 (SHOWA DENKO K.K.) refractive index detector. Wyatt Technologies software (ASTRA) was used to determine the corresponding peaks' molecular weight based on the refractive index.

**Deubiquitination assays on a minimal substrate.** Enzyme activity of USP7 was measured using the fluorescence of rhodamine upon cleavage of the quenched minimal substrate UbRho (Ubiquitin-Rhodamine110Gly, Ub-Rh110Gly; UbiQ, the Netherlands). Experiments were performed in running buffer (20 mM HEPES, pH 7.5, 100 mM NaCl, 1 mM DTT, 1 mM EDTA and 0.05% v/v Tween-20). Protein samples were prepared at 2X concentration and added to 8 μM UbRho just before measuring, reaching an end volume of 20 μL in the plate. The release of rhodamine was measured at the emission wavelength of 520 nm (±10 nm) after excitation at 485 nm (±10 nm) in a Pherastar plate reader (BMG LABTECH GmbH, Germany). Either the raw data were plotted directly in Prism 7 (GraphPad), or the slopes were converted to initial velocity values for plotting against the titration range. Assays were performed three times with two different protein batches.

For steady-state kinetics analysis a single concentration of USP7 constructs (CD, CD12345 and CD12345$^{\Delta C}$) was incubated with a dilution range of UbRho and assessed for activity using the same experimental setup as described above. The initial velocities were determined using the linear slope of the reaction and plotted against the concentration UbRho used. Using Prism 7 the data were fitted using the Michaelis-Menten equation, yielding the reported steady-state kinetics parameters.

For kinetics analysis in KinTek, a concentration series of UbRho was used with USP7 constructs FL, (1 nM), TCD and CD (both 20 nM). To get resolution at the earliest time points the assay was performed using the injector, injecting the enzyme into the UbRho solution followed by direct detection (as described above). The resulting values were converted to rhodamine concentrations before being loaded into KinTek.

**NMR experiments.** All NMR experiments were carried out on Bruker Avance III HD spectrometer operating at 850 MHz $^1H$ Larmor frequency and equipped with a

cryoprobe. All NMR spectra were processed using Bruker TopSpin or NMRPipe[60]. NMR samples for assignment contained 180 µM USP7-Ubl45 with either uniform $^1$H, $^{15}$N, $^{13}$C or fractional $^2$H, uniform $^{15}$N, $^{13}$C labelling in 50 mM HEPES, pH 7.5, 100 mM NaCl, 7% $D_2O$ and 1 mM DTT (NMR buffer). Backbone resonances of Ubl45 were assigned to 84% completeness, using 3D TROSY HNCO, HN(CA)CO, HNCA, HNCOCA, HNCACB, HNCOCB, CBCA(CO)NH spectra. Assignment was done using CCPN[61]. The program TALOS[62] was used to analyse the secondary structure based on the assigned backbone chemical shifts.

Titration of $^1$H$^{15}$N-labelled Ubl45 (45 µM) with either USP7CD (using Ubl45: CD molar ratios of 1:0, 1:1, 1:5 or 1:10) or USP7CDUb (using ratios of 1:0.1 and 1:0.3) were performed after extensive dialysis of the proteins to NMR buffer. We monitored residue-specific intensity change and chemical shift perturbations (CSP) of Ubl45 amide backbone resonances in 2D $^1$H$^{15}$N TROSY spectra. The CSPs were calculated from the perturbations in the $^1$H ($\Delta\delta_H$) and $^{15}$N ($\Delta\delta_N$) dimensions as the weighted average (composite) CSP in ppm according to ref. [63]. The intensity changes were plotted against residue number for the end point of both titrations.

**Synthesis of p53-conjugated ubiquitin reagents.** Both ubiquitin and the C-terminus of p53 were produced synthetically by solid phase peptide synthesis, for the native reagent the p53 peptide was N-terminally labelled with 5-carboxytetramethylrhodamine (TAMRA). The peptide was linked to ubiquitin using click chemistry or native chemical ligation[49] to yield the non-hydrolysable[48], natively linked[44] and covalently binding[45] p53Ub and p53$_{short}$Ub probes. Details are available in  Supplemental methods section.

**Generation of covalent ubiquitin-USP7 complexes.** A total of 120 µM CD was incubated overnight with an excess of Ub-PA[64] and Ubl45 (both 150 µM) under dialysis (against 20 mM HEPES, pH 7.5, 50 mM NaCl, 2 mM DTT), yielding 80% of the CD reacted with the ubiquitin probe. The sample was subjected to anion exchange (PorosXQ) and gel filtration (Superdex 75) to remove unreacted ubiquitin and Ubl45. Fractions were concentrated for use in affinity assays. For USP7-TCD a similar approach was used, only substituting p53Ub$_{VA}$ for Ub-PA, resulting in 100% reaction.

**Surface plasmon resonance assays.** All surface plasmon resonance (SPR) experiments were carried out on a Biacore T200 machine (GE Healthcare) at 25 °C. A polyclonal GST antibody from the GST capture Kit (GE Healthcare) was covalently bound on a CM5 sensor chip via amino coupling. Two-hundred units of GST-tagged USP7 constructs were immobilised on the test flow cell, whilst the blank flow cell had an equal amount of GST only immobilised. The C-terminal peptide (residues 1083–1102) was synthesized with a biotin at the N-terminus and immobilized up to ~30 RU on a SA chip. A concentration series of USP7 constructs with or without covalently bound ubiquitin probe was tested for binding using running buffer (20 mM HEPES, pH 7.5, 100 mM NaCl, 1 mM DTT, 1 mM EDTA and 0.05% v/v Tween-20) supplemented with 1 mg mL$^{-1}$ BSA and 1 mg mL$^{-1}$ dextran.

Interaction values ($K_D$) were determined by plotting steady-state equilibrium values against the concentration and fitting these with 1:1 stoichiometry using Prism 7 (Graphpad). For easy comparison purpose, responses were normalised using B$_{max}$. For binding curves with detectable dissociation and a $K_D$ below 10 µM we used EvilFit[42] to determine kinetic rate constants. All experiments were performed at least *in duplo* and representative curves are shown.

**Fluorescence polarization binding assays.** To measure the affinity for ubiquitin, N-terminally tetramethylrhodamine (TAMRA) labelled was incubated with a titration range of each USP7 construct. All assays were performed in running buffer (20 mM HEPES, pH 7.5, 100 mM NaCl, 1 mM DTT, 1 mM EDTA and 0.05% v/v Tween-20) on a Pherastar plate reader (BMG LABTECH GmbH, Germany), using excitation wavelength 540 nm (±20 nm) and detection of polarization at 590 nm (±20 nm). The anisotropy of $^{TAMRA}$Ub was calibrated at 35 mA, any change in anisotropy upon USP7 interaction was calculated using MARS data analysis software (BMG LABTECH GmbH, Germany) and plotted using Prism 7 (GraphPad).

The affinity between the USP7 constructs and a TAMRA-labelled p53 peptide ($^{TAMRA}$p53) was measured on a ClarioStar plate reader (BMG LABTECH GmbH, Germany). Assays were performed *in triplo* using the same running buffer and wavelength filters (Ex. 540 ± 20 nm, Em. 590 ± 20 nm). $^{TAMRA}$p53 anisotropy was calibrated to be 35 mA and changes in anisotropy were plotted and fitted in Prism 7 to obtain affinities.

The p53 FP binding assay was repeated for USP7FL in a stopped-flow setup. 25 nM of $^{TAMRA}$p53 was incubated with a concentration range of USP7 in running buffer. The binding was monitored using an excitation wavelength of 548 nm on a TgK Scientific instrument (model SF-61DX2) equipped with photomultiplier tube R10699 (Hamamatsu) and Kinetic Studio was used to merge ten separate, sequential injections for each protein concentration.

**USP7 inhibition assays.** USP7 constructs were incubated for 30 min in assay buffer (50 mM Tris-HCl, pH 7.6, 100 mM NaCl, 2 mM cysteine, 1 mg mL$^{-1}$ CHAPS) with various concentrations of the non-hydrolysable p53Ub$_{inh}$ construct,

prior to assessment in a deubiquitination assay. To account for difference in activity, protein concentrations were adapted for FL (0.35 nM), CD12345 (0.35 nM), TCD (200 nM) and CD (75 nM), whilst the substrate (UbRho) concentration was kept constant at 0.4 µM. Protein samples and substrate were prepared at 4× the final concentration. The initial raw velocities were derived and plotted against the titration range of inhibitor reagent. Using Prism 7 the data were fitted to yield IC$_{50}$-values.

**Fluorescence polarization activity assays.** Various USP7 constructs at indicated concentrations were incubated with 100 nM of TAMRA-labelled, natively linked p53Ub ($^{TAMRA}$p53Ub) to trace the binding and hydrolysis of the reagent. Assays were performed in assay buffer (20 mM Tris-HCl, pH 7.6, 100 mM NaCl, 1 mM DTT and 1 mg mL$^{-1}$ CHAPS) on a Pherastar plate reader measuring at 590 nm after excitation at 540 nm. The FP signal for $^{TAMRA}$p53Ub only was used as a starting baseline, whilst the TAMRA-labelled p53-peptide represents the fully cleaved reagent.

The fluorescence polarization activity assays were repeated in a stopped-flow setup. USP7 constructs at three concentrations (50 nM, 25 nM and 12.5 nM) were incubated with 50 nM of $^{TAMRA}$p53Ub to trace the binding and hydrolysis of the reagent. $^{TAMRA}$p53Ub was monitored in running buffer (20 mM HEPES, pH 7.5, 150 mM NaCl, 1 mM DTT and 0.05% v/v Tween-20) using an excitation wavelength of 548 nm on a TgK Scientific instrument (model SF-61DX2) equipped with photomultiplier tube R10699 (Hamamatsu), the manufacturer's software (Kinetic Studio) was used to merge the ten measurements performed for each concentration.

**KinTek modelling.** All data used were imported into KinTek with concentrations in µM and time in seconds: for minimal substrate activity curves, converted to released rhodamine, could be loaded into KinTek directly. The curves resulting from USP7 inhibition assays were read with a delay time of 120 s. The FP activity assay data from the stopped-flow instrument could also be read directly. With the data for every construct imported, the model (Fig. 6h) was fitted per construct separately for each experiment. When the fits proved stable, reaction constants were linked and a global fit was performed. The resulting values were then statistically tested using the FitSpace module of the Kintek software. For detailed information see Supplementary Methods.

**Reporting summary.** Further information on experimental design is available in the Nature Research Reporting Summary linked to this article.

## Data availability
NMR assignments for Ubl45 (residues 890–1102) are deposited in the Biomolecular Magnetic Resonance Bank (BMRB 27627). Other datasets generated during and/or analysed during the current study are available from the corresponding author on reasonable request. The source data underlying Figs. 1–6 and Supplementary Figures 1–7 are provided as a Source Data file. A reporting summary for this article is available as a Supplementary Information file.

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

## Acknowledgements

We thank an anonymous reviewer for helpful advice on the NMR analysis. The authors thank Shreya Dharadhar, Robbie Joosten, Anastassis Perrakis and Michael Uckelmann for critical reading of the manuscript. Research was supported by KWF (2012–5398), NWO TOP grant (TOP 714.016.002) and the Oncode institute to T.K.S., NWO VIDI grant (723.013.010) to H.v.I and NWO VICI grant (724.013.002) to H.O.

## Author contributions

R.Q.K. designed, performed and analysed experiments and wrote the manuscript, W.J.v. D and R.Q.K. expressed and purified proteins. P.P.G., R.E. and M.P.C.M. designed and synthesised a p53Ub toolbox, performed inhibition experiments and initial validation of the synthesised ubiquitin reagents. F.E.O. and D.v.D. performed ubiquitin and peptide synthesis. F.E.O. contributed to peptide and chemical design and provision of reagents. A.F. and R.Q.K. performed SPR and stopped-flow, their analysis and global modelling.

H.v.I. and R.Q.K. performed NMR experiments and analysis. H.O. supervised synthetic ubiquitin research. T.K.S. supervised and designed research and wrote the manuscript. All authors read the manuscript critically.

## Additional information

**Competing interests:** F.E.O. is a current employee, co-founder and shareholder of UbiQ Bio BV. H. O. is co-founder, and shareholder of UbiQ Bio BV. T.K.S. is member of the scientific advisory board of Mission Therapeutics. The remaining authors declare no competing interests.

