## [Peer Review File · Nature Communications]

Reviewers' comments:

Reviewer #1 (Remarks to the Author):

This manuscript by Kim et al describes an interesting study of Usp7 deubiquitinase (DUB) targeted at unraveling mechanistic steps during deubiquitination catalyzed by the DUB, which represents a striking example of catalytic regulation that combines a complex set of autoregulatory and substrate-induced effects. The authors present an innovative mix of classical biochemistry (enzymology), biophysical chemistry and application of novel chemical biology tools to dissect steps during USP7 catalysis. The paper is clearly written with nice illustrations, and the data are solid with consistent interpretation. A number of interesting observations were made resulting in delineation of sequence of activation events that the enzyme seems to undergo to reach a catalytically competent state. It is hard to disagree with most of their arguments, which I think are duly supported by data. I would be delighted to see this story published in a journal of highest profile.

It is clear that the authors have been very thoughtful in their design of experiments and work appears to be quite comprehensive. I do think some more clarity is required to make this already good paper better accessible to readers. Please consider the following points, which I hope may improve the manuscript.

(1) I think the kinetic results should be better explained (Figure 6 and associated results). In my view, this is one of the most innovative elements in the report. The authors have synthesized novel substrates (ubiquitinated p53 peptides) and applied them to extract rate constants through stopped-flow technique. The resulting data are used as inputs in a theoretical modeling of kinetic steps. I urge the authors to include more discussion in the results section of the main body of the text (the explanation of how the kinetic data are used to obtain rate parameters are placed in supporting information. I think bringing some of that to the main text would help.) It is somewhat unclear how the raw data (stopped-flow data) was used to extract rate constants of different steps. For example, at one point the authors seem to indicate that the part of the stopped-flow profile in Figure 6c corresponding to reduction in fluorescence polarization (FP) is indicative of hydrolysis, which I agree. It appears to me that this segment of the profile is interpreted to include two distinct steps, release of ubiquitin and the p53 peptide after hydrolysis. It is not clear to this reader how that is so.

Please indicate which part of the substrate in figure 6C is labeled with the fluorescent reporter. I assume that it is on the peptide part.

Will it be better to use k_f and k_r (forward and reverse rate constants) instead of k_+ and k_- in Figure 6H? Please include units.

(2) The section on a second p53 interaction site needs further clarification. Is it a different site, meaning two sites for substrate binding? Or, an extended interface for substrate binding with contribution from Ubl domains?

Reviewer #2 (Remarks to the Author):

The manuscript by Kim and coauthors describes a comprehensive kinetic analysis of USP7 activation mechanism using SPR and stopped flow in combination with chemically synthesized mimetics of USP7 substrate. Authors confirm previously described activating effect of the USP7 C-terminus and show that substrate binding may contribute to the activation mechanism. The study is thorough, well

executed and provides important insight into the mechanism of USP7 catalysis. The major innovation of the study is the proposed multistep reaction kinetic modelling that implies that USP7 target recognition is an important driver of the reaction. These findings are of general interest to ubiquitination community and enzymology field.

Overall, this is a very interesting manuscript that is suitable for publication in Nature Communications with minor revisions. Please see my comments below:

The fact that C-terminus of USP7 is critical and can activate the catalytic domain of the enzyme is not novel (Ma et al, Arch. Biochem. Biophys., 2010; Faesen et al, Mol. Cell, 2011); however, the fact that it happens in cis rather than in trans as was previously captured in crystal structure of the Ubiquitin-Catalytic-UBL45 chimera (Rougé et al, Structure, 2016) is novel and represents an important contribution to the field.

The question remains how exactly do the C-terminus and UBL45 bind the catalytic domain. Based on their NMR studies the authors imply that the crystal structure is an artifact; however, they provide no alternative structural model for the complex formation. Furthermore, the NMR chemical shift perturbations presented in the manuscript are minimal and not convincing due to extremely weak binding affinities.

Finally, no other NMR studies of USP7 are referenced in this manuscript, including the solution studies of the interaction between USP7 catalytic domain and ubiquitin. A reference to previous NMR studies showing that free ubiquitin does not affect the conformation of the USP7 active site and needs to be conjugated to a substrate to do so (Pozhidaeva et al, Cell Chem. Bio, 2017) would strengthen the authors major claim that substrate recognition plays an important role in USP7 activation.

Reviewer #3 (Remarks to the Author):

The manuscript "Kinetic analysis of multi-step USP7 mechanism shows critical role for target protein in activity" by Kim et al. investigates how USP7 is activated by a multi-step process, involving both target and ubiquitinated substrate binding, to align the USP7 catalytic triad from an inactive, apo conformation into a catalytically active protease. This work is of interest as it may provide a framework to understand how ubiquitinated substrates collaborate with deubiquitinases facilitate their own proteolytic cleavage. The use of multiple biophysical methods to cross-validate data is a strength of the manuscript, however in some cases (described below) the different methodologies require additional experimental detail to provide an internally-consistent mechanism. Overall the studies may be acceptable for publication in Nature Communications, provided that the points below are sufficiently addressed.

Major points

Lines 88-90 (and throughout the manuscript): the authors focus on one model of the Rougé et al. crystal structure, suggesting that trans-activation was the only proposed mechanism. The Rougé et al. manuscript points just as much to a cis-mechanism: "it is possible that cis and/or trans activation could occur depending on the nature of the substrate". Furthermore the graphical abstract of the Rougé et al. manuscript depicts a model of cis-activation, indicating that trans-activation is not a central hypothesis. The authors should re-word their manuscript to present a more balanced representation of the Rougé et al. manuscript.

line 116: the protein concentration in a size exclusion chromatography run is used to estimate

presence of (transient) dimer. Can the authors provide an estimate on the protein concentration as the eluate passes the SEC-MALS detector, for example by a UV detection method? By the time the protein passes the detector, it may well be very diluted.

lines 130-140: it is possible that the shift data of labeled Ubl45 can be interpreted differently than as presented. The chemical shift perturbations may also be an indication that a ternary complex of two Ubl45 and two CDs is forming. Even though the shifts do not recapitulate the x-ray structure of Rougé et al. they also do not unequivocally show that there is not a dimer. On the contrary, the lack of perturbations but abundance of line broadening could indicate the formation of a high molecular weight complex. The statement that this is indicative of multiple binding modes is therefore speculative and not necessarily represented by the data.

Fig. 2, Fig. S1: major perturbations are observed for histidine residues, that are extremely sensitive to pH changes. Is it possible that the observed changes are the result of slight experimental errors in the pH determinations? The next largest changes are in Glu and Asn residues, that are also pH-sensitive groups. Collectively, these residues represent half of the observed shifts and the remaining shifts are rather small. Additionally, the postulation that at nM concentrations, the probability of a dimerization is reasonable. However the NMR shift mapping studies are done at orders of magnitude higher concentrations; thus the dimerization postulate is speculative with the current dataset. Can data be acquired to show the concentration dependence and then be extrapolated to nM concentrations?

Fig. 2 implies that the tail region of Ubl45 interacts with the unlabeled USP7-CD. Yet the authors state that the lack of loss of line intensity implies there is no interaction (line 140, Fig, 2b, 2c). In principle, the NMR data are contradictory as line broadening assumes intermediate- to high-affinity interactions yet the minimal shift perturbations indicate only transient, non selective binding. These findings require additional clarification.

Line 144: the minimal chemical shift perturbations do not explain the observed line broadening effects and that this contradiction should be experimentally addressed. One path could be to do titration experiments and determine the binding Kd based on line broadening and compare that to the Kd based on chemical shift perturbations. In this way, the effect of direct interaction (as perhaps by an encounter complex) may be separated from the induced fit formation of a higher molecular weight complex that may well be a dimer.

Line 145: the observations and conclusions are unclear. The chemical shift perturbations in the tail are the second largest observed, do these not indicate that the C-terminal tail is interacting with the CD?

Fig. 3a: the tailless Ubl45 interaction is determined by SPR to be 420uM (line 155) and Ubl45 with tail is determined to be 280uM, it is therefore surprising that the NMR experiment does not show more shift perturbations. Protein-protein interactions typically involve multiple amino acids and induce many chemical shift changes. At a 10:1 USP7-CD/Ubl45 ratio, the complex at the reported Kd values is expected as be the majority of the species in solution, thus it is unclear why there very few shifts and also very small perturbations are observed. Do the authors have an explanation?

line 167: the active conformation of USP7 is only visible when substrate is covalently attached to the active site cysteine using Ub-PA, that is an unnatural ubiquitin derivative. This experiment does not necessarily indicate that the same stabilization of the active site triad occurs by Ubiquitin binding in a non-covalent fashion. Can the authors demonstrate that Ub binding to CD alone is enough to facilitate the active conformation upon addition of Ubl45? This may be achieved by repeating the NMR experiments in the presence of high concentrations of unlabeled ubiquitin.

Line 193: the hypotheses that Ubl45 Δ C promotes ubiquitin binding could be further supported by NMR data. The affinity of labeled ubiquitin (or p53 peptide-conjugated labeled ubiquitin) could be evaluated with Ubl45 Δ C, and Ubl45 with tail could be added for more clarity.

The authors generate mono-ubiquitinated p53-peptide reagents. Physiologically, p53 is ubiquitinated with polyubiquitin chains. How does the model change when substrates are modified with polyubiquitin chains, particularly when the polyubiquitin chains have different linkage types? How does the model change when p53 is modified at the other five ubiquitination sites on the C-terminus?

Lines 255-256: The authors should generate USP7 mutants in key TRAF and Ubl12345 residues implicated to participate in the binding of ubiquitinated-p53 in order to more precisely evaluate the proposed avidity effects.

Minor point:

Line 53: USP7 mutations are correlated with pediatric hematologic tumors, but there are no data to indicate these mutations are causative and drive these malignancies.

We thank the reviewers for their positive and constructive comments. To address their comments we have added the following experiments to the manuscript:

- Better mapping of the secondary p53 interaction on USP7 using additional fluorescence polarization assays to analyse binding of the peptide to a series of different USP7 constructs (**new Fig. 5A**).
- A steady-state kinetic analysis of USP7CD12345^{ΔC} and a comparison to other constructs, to strengthen the finding that in the absence of the C-terminal tail, Ubl45^{ΔC} still enhances the affinity of CD for ubiquitin (**new Fig. 3G**).
- A repeat of the Ubl45-CD NMR titration, with extensive and simultaneous dialysis to rule out pH effects. The new data reproduces the generic loss in peak intensity and the lack of residue-specific effects, indicating a weak, non-specific interaction between Ubl45 and CD, predominantly mediated by the Ubl45 core and not the C-terminal tail (**updated Fig. 2**).
- An in-depth quantitative analysis and simulation of NMR data to clarify our claim that Ubl45 and CD interact in weak, dynamic, multi-state manner in absence of ubiquitin. (**new Fig. S2**).
- An additional NMR titration experiment, adding CD with conjugated ubiquitin to Ubl45, which shows that only in presence of ubiquitin the tail is immobilized in the complex (**new Fig 2E**).

Additional new experiments are shown below in the point by point response to reviewers. A version of the manuscript with textual changes highlighted has been added.

Reviewer 1

I think the kinetic results should be better explained (Figure 6 and associated results). In my view, this is one of the most innovative elements in the report. The authors have synthesized novel substrates (ubiquitinated p53 peptides) and applied them to extract rate constants through stopped-flow technique. The resulting data are used as inputs in a theoretical modeling of kinetic steps. I urge the authors to include more discussion in the results section of the main body of the text (the explanation of how the kinetic data are used to obtain rate parameters are placed in supporting information. I think bringing some of that to the main text would help.)

We thank the reviewer for the enthusiasm for the analysis. We have followed the suggestion and add more detail to the description of the process of modelling in the main text.

It is somewhat unclear how the raw data (stopped-flow data) was used to extract rate constants of different steps. For example, at one point the authors seem to indicate that the part of the stopped-flow profile in Figure 6c corresponding to reduction in fluorescence polarization (FP) is indicative of hydrolysis, which I agree. It appears to me that this segment of the profile is interpreted to include two distinct steps, release of ubiquitin and the p53 peptide after hydrolysis. It is not clear to this reader how that is so.

We have provided a more extensive explanation that these steps could not be modelled in a single step. The different affinities for the two products, Ub and the free peptide, suggest a specific order of release but the modelling results are not affected by changing this order.

Please indicate which part of the substrate in figure 6C is labeled with the fluorescent reporter. I assume that it is on the peptide part.

Indeed, it is on the first amino acid of the peptide. We have changed figure 6C (and related supplemental figures) and its legend to make this point more explicit.

Will it be better to use k_f and k_r (forward and reverse rate constants) instead of k_+ and k_- in Figure 6H? Please include units.

We have implemented this suggestion in the text and figures.

(2) The section on a second p53 interaction site needs further clarification. Is it a different site, meaning two sites for substrate binding? Or, an extended interface for substrate binding with contribution from Ubl domains?

To map the interaction we have added new FP data measuring interaction of the peptide with all USP7 constructs. We could show this binding requires the presence of Ubl45, but not its tail (new Fig 5a). We added this mapping to the main text.

Reviewer #2 (Remarks to the Author):

The fact that C-terminus of USP7 is critical and can activate the catalytic domain of the enzyme is not novel (Ma et al, Arch. Biochem. Biophys., 2010; Faesen et al, Mol. Cell, 2011); however, the fact that it happens in cis rather than in trans as was previously captured in crystal structure of the Ubiquitin-Catalytic-UBL45 chimera (Rougé et al, Structure, 2016) is novel and represents an important contribution to the field. The question remains how exactly do the C-terminus and UBL45 bind the catalytic domain. Based on their NMR studies the authors imply that the crystal structure is an artifact; however, they provide no alternative structural model for the complex formation. Furthermore, the NMR chemical shift perturbations presented in the manuscript are minimal and not convincing due to extremely weak binding affinities.

The referee rightly points out the challenge for the NMR study of the Ubl45-CD interaction. The concerns of this reviewer and reviewer 3 have prompted us to redo this experiment and the new data is presented in the revised Figure 2 together with a titration of Ubl45 with CD~Ub. The data imply that in absence of Ub, Ubl45 and CD interact differently than presented in the crystal structure: the Ubl45 core interacts weakly with CD without immobilization of the tail. Because of the absence of residue-specific effects an interface cannot be defined and modelling of the complex is thus impossible.

In presence of Ub, which corresponds closely to the crystallized construct, the interaction is much tighter and immobilizes the tail. Since this interaction is in the slow exchange regime and the bound state cannot be observed directly due to its size (75kDa), no residue-specific effects are observed. Thus we cannot verify or refute the binding mode observed in the crystal.

It should also be noted that given the size of the Ubl45-CD complex (67 kDa), the observation of only minimal CSPs in the CD titration is to be expected. Residues with large chemical shift changes would not show large CSPs but rather a strong peak intensity decrease (see the new analysis in Supplemental Figure S2).

We have added a rigorous analysis and simulation of the NMR titration experiment in Supplemental Figure S2 and changed the main text to include the results.

Finally, no other NMR studies of USP7 are referenced in this manuscript, including the solution studies of the interaction between USP7 catalytic domain and ubiquitin. A reference to previous NMR studies showing that free ubiquitin does not affect the conformation of the USP7 active site and needs to be conjugated to a substrate to do so (Pozhidaeva et al, Cell Chem. Bio, 2017) would strengthen the authors major claim that substrate recognition plays an important role in USP7 activation.

We apologize for this omission and have added the reference and a discussion of its results.

Reviewer #3 (Remarks to the Author):

Major points

Lines 88-90 (and throughout the manuscript): the authors focus on one model of the Rougé et al. crystal structure, suggesting that trans-activation was the only proposed mechanism. The Rougé et al. manuscript points just as much to a cis-mechanism: "it is possible that cis and/or trans activation could occur depending on the nature of the substrate". Furthermore the graphical abstract of the Rougé et al. manuscript depicts a model of cis-activation, indicating that trans-activation is not a central hypothesis. The authors should re-word their manuscript to present a more balanced representation of the Rougé et al. manuscript.

This is correct, we apologize and we have changed our wording to adjust the text.

line 116: the protein concentration in a size exclusion chromatography run is used to estimate presence of (transient) dimer. Can the authors provide an estimate on the protein concentration as the eluate passes the SEC-MALS detector, for example by a UV detection method? By the time the protein passes the detector, it may well be very diluted.

Indeed, the protein passes the detector diluted. We now mention elution concentrations explicitly in the text.

lines 130-140: it is possible that the shift data of labeled Ubl45 can be interpreted differently than as presented. The chemical shift perturbations may also be an indication that a ternary complex of two Ubl45 and two CDs is forming. Even though the shifts do not recapitulate the x-ray structure of Rougé et al. they also do not unequivocally show that there is not a dimer. On the contrary, the lack of perturbations but abundance of line broadening could indicate the formation of a high molecular weight complex. The statement that this is indicative of multiple binding modes is therefore speculative and not necessarily represented by the data.

The concerns of this reviewer and reviewer 2 regarding the titration have prompted us to redo this experiment. The new data, obtained after extensive simultaneous dialysis of the proteins, reproduces the extensive peak intensity loss and the lack of residue-specific effects. We note that the observed peak intensity decrease is in good agreement with the formation of a 67 kDa complex. Formation of a higher-molecular weight complex (134 kDa for a dimer of dimers) would result in much lower observed peak intensities for Ubl45 core residues at the end of our titration, for reasons explained in Figure S2, panel A. Combined with monomeric behaviour of

the Ubl45 and CD domains at NMR concentrations (Figure R2 in this rebuttal), and the weak Ubl45-CD affinity we favor the interpretation of a weak, dynamic multi-state complex. Nevertheless, alternative scenarios cannot be formally excluded. We have adjusted the text to make our reasoning clearer.

Fig. 2, Fig. S1: major perturbations are observed for histidine residues, that are extremely sensitive to pH changes. Is it possible that the observed changes are the result of slight experimental errors in the pH determinations? The next largest changes are in Glu and Asn residues, that are also pH-sensitive groups. Collectively, these residues represent half of the observed shifts and the remaining shifts are rather small.

We thank the reviewer for this comment. We analyzed Ubl45 spectra at pH 7.5 and 5.5 and found that most CSPs are close to what could be expected from a pH change. We therefore performed the experiment again after simultaneous and extensive dialysis of the proteins. As mentioned above, the new data reproduces the generic peak intensity loss, but this time no significant CSPs are observed, except for a residue in the tail.

Additionally, the postulation that at nM concentrations, the probability of a dimerization is reasonable. However the NMR shift mapping studies are done at orders of magnitude higher concentrations; thus the dimerization postulate is speculative with the current dataset. Can data be acquired to show the concentration dependence and then be extrapolated to nM concentrations?

The dimerization of the full-length protein in which CD and Ubl45 are linked does not occur at least up to concentrations of 20 μ M (Fig. 1). We perform our NMR experiments using separated CD and Ubl45. We have analysed these constructs on size exclusion chromatography at equivalent concentrations (in the elution) and compared the elution profile to a calibration with globular proteins. This leads us to conclude that these constructs elute as monomers.

Figure R2. Both USP7-CD and USP7-Ubl45 were run on a Superdex75 gel filtration column. Both eluted as a monomer with peak concentrations of $\sim 70 \mu\text{M}$ (for Ubl45; 25 kDa) and $\sim 75 \mu\text{M}$ (for CD; 42 kDa). Injection concentrations were about 7 times higher for both, for CD the detector limit was reached.

Fig. 2 implies that the tail region of Ubl45 interacts with the unlabeled USP7-CD. Yet the authors state that the lack of loss of line intensity implies there is no interaction (line 140, Fig, 2b, 2c). In principle, the NMR data are contradictory as line broadening assumes intermediate- to high-affinity interactions yet the minimal shift perturbations indicate only transient, non selective binding. These findings require additional clarification.

Indeed our description was too concise, for which we apologise. In the new pH-controlled data, we see only very minor CSPs for the tail and little intensity changes, indicating the tail is not stably associated with the CD domain. The large intensity drop for the core indicated the interaction is predominantly mediated via the Ubl45 core. However, upon addition of CD~Ub, the tail and core residues show similar intensity losses, indicating immobilization of the tail on the CD~Ub surface. We have rephrased the text to better describe the observations and our interpretation.

Line 144: the minimal chemical shift perturbations do not explain the observed line broadening effects and that this contradiction should be experimentally addressed. One path could be to do titration experiments and determine the binding K_d based on line broadening and compare that to the K_d based on chemical shift perturbations. In this way, the effect of direct interaction (as perhaps by an encounter complex) may be separated from the induced fit formation of a higher molecular weight complex that may well be a dimer.

The combination of very small CSPs and large line broadening may indeed seem contradictory. Again our description was too concise to explain that this combination is exactly to be expected upon formation of a 67 kDa complex. We have included an

extensive quantitative analysis of the NMR experiment, including simulation experiments, in Figure S2 to show that (i) the observed line broadening matches quantitatively with that expected from the SPR-determined K_D ; (ii) only small CSPs can be observed as residues with larger CSPs will suffer from excessive line broadening preventing their observation. Thus, any specific interaction site would be recognizable from a specific loss of peak intensity beyond the overall generic intensity loss, rather than large CSPs.

Accurate determination of the K_D beyond the analysis provided in Figure S2, is unfortunately not possible here, since saturation of the Ubl45 domain would require ~6 mM CD, which is not feasible. Furthermore, both off-rate and K_D determine the peak intensity for the core residues, but are highly correlated. As shown in Figure S2a, the data match well with SPR-derived K_D , but it could also fit to higher K_D values with faster exchange.

Our model of a weak, dynamic Ubl45-CD complex could be seen as an encounter-complex. We suggest that the presence of ubiquitinated target is required to induce the specific binding mode of the tail.

We have rephrased the text to better explain what we observe and included an extensive analysis and simulation experiments to explain this in detail.

Line 145: the observations and conclusions are unclear. The chemical shift perturbations in the tail are the second largest observed, do these not indicate that the C-terminal tail is interacting with the CD?

As mentioned above, the new data indicates the tail is not in the 'locked' state that would be expected from the peptide binding in the crystal structure of Rougé et al. when titrating with CD only. Addition of CD~Ub does result in immobilization of the tail. We have changed the text to better explain this point.

Fig. 3a: the tailless Ubl45 interaction is determined by SPR to be 420uM (line 155) and Ubl45 with tail is determined to be 280uM, it is therefore surprising that the NMR experiment does not show more shift perturbations. Protein-protein interactions typically involve multiple amino acids and induce many chemical shift changes. At a 10:1 USP7-CD/Ubl45 ratio, the complex at the reported K_D values is expected as be the majority of the species in solution, thus it is unclear why there very few shifts and also very small perturbations are observed. Do the authors have an explanation?

Indeed, we calculate ~60% occupancy of the complex under these conditions We describe in Figure S2 why there are no clear CSPs: given the size of the complex and the use of protonated protein, the bound state is largely invisible. Residues with large chemical shift changes in a weak affinity interaction, as between Ubl45 and CD, will show with a specific loss of intensity instead. As mentioned above, because of the absence of such residue-specific effects we conclude that the interaction is more like a series of transient interactions rather than a defined high affinity interaction. We have extended our explanation of this in the text.

line 167: the active conformation of USP7 is only visible when substrate is covalently attached to the active site cysteine using Ub-PA, that is an unnatural ubiquitin derivative. This experiment does not necessarily indicate that the same stabilization of the active site triad occurs by Ubiquitin binding in a non-covalent fashion. Can the authors demonstrate

that Ub binding to CD alone is enough to facilitate the active conformation upon addition of Ubl45? This may be achieved by repeating the NMR experiments in the presence of high concentrations of unlabeled ubiquitin.

This was extensively studied by Pozhidaeva et al, whose paper we accidentally omitted in our manuscript (see comments to reviewer 2). They show that a ubiquitin-substrate will affect the CD conformation, but not free ubiquitin (the product of the reaction).

Line 193: the hypotheses that Ubl45 Δ C promotes ubiquitin binding could be further supported by NMR data. The affinity of labeled ubiquitin (or p53 peptide-conjugated labeled ubiquitin) could be evaluated with Ubl45 Δ C, and Ubl45 with tail could be added for more clarity.

We have addressed this point in a different way, using Michaelis-Menten analysis of the enzyme constructs (new Fig. 3g). When we study activity of CD12345 Δ C on a minimal substrate (UbRho) we observe that it has the same K_M as full length USP7 (~5 μ M) rather than that of CD (19 μ M). Of course the k_{cat} is much lower for this construct (Fig. 3g). We have adjusted the text to explain this point.

The authors generate mono-ubiquitinated p53-peptide reagents. Physiologically, p53 is ubiquitinated with polyubiquitin chains. How does the model change when substrates are modified with polyubiquitin chains, particularly when the polyubiquitin chains have different linkage types? How does the model change when p53 is modified at the other five ubiquitination sites on the C-terminus?

We and others have previously shown that USP7 is active on a variety of chain types and substrates. Although we expect that avidity between TRAF and the rest of USP7 are retained, the detailed numbers will be different on these substrates. To perform quantitative data on such a substrate is beyond the scope of this manuscript.

Lines 255-256: The authors should generate USP7 mutants in key TRAF and Ubl12345 residues implicated to participate in the binding of ubiquitinated-p53 in order to more precisely evaluate the proposed avidity effects.

We have shown the presence of the avidity between TRAF and Ubl12345 in several different ways: the binding experiment with the covalent probe (Fig. 4c), the FP binding experiment that we have now expanded (Fig. 5a), inhibition with non-cleavable substrate (Fig. 5b) and stopped-flow analysis on a cleavable substrate, using 4 different constructs. Since all these orthogonal data agree we feel that the presence of avidity is sufficiently robust and further analysis would not contribute to understanding. Of course we would love to have a high resolution structure of the complex, but we feel that this is beyond the scope of this manuscript

Minor point:

Line 53: USP7 mutations are correlated with pediatric hematologic tumors, but there are no data to indicate these mutations are causative and drive these malignancies.

We have adjusted the text to reflect that this is not formally proven.

Reviewers' comments:

Reviewer #1 (Remarks to the Author):

My concerns have been adequately addressed. I see that the current version is improved with better clarity, not just in the kinetic results but also in overall presentation.

Reviewer #2 (Remarks to the Author):

The revised manuscript fully addresses my previous comments. The new set of NMR experiments shows no chemical shift perturbations in UBL45 spectrum upon the CD binding and moderate signal broadening, which authors attribute to a non-specific weak binding between the domains. The UBL45 binding to the CD-Ub, on the other hand, causes more dramatic signal broadening, suggesting a tighter binding in the presence of ubiquitin, in agreement with their other binding experiments. The NMR data showing that the C-terminus of the UBL45 becomes more ordered, however, is not conclusive because the peak broadening upon the CD-Ub binding is quite uniform and affects the C-terminus the same way it affects the N-terminus, unless both, the N- and the C-termini now interact. Apart from a few inconsistencies that can be attributed to the limitations of the methods used, the paper is interesting and warrants publication.

Reviewer #3 (Remarks to the Author):

Most of the technical comments from reviewers are adequately addressed. Two significant issues remain:

- 1) The authors continue to focus on one model of the crystal structure reported in Rougé et al (Structure 2016, <http://dx.doi.org/10.1016/j.str.2016.05.020>) and indicate that trans-activation is the only proposed mechanism. This limited interpretation is inappropriate as worded in the current form of the Kim et al. manuscript.
- 2) Data discrepancies and lack of direct comparisons between the constructs used in the Kim et al and Rougé et al manuscripts, that should be resolved.

Issue #1: Misrepresentation of the Rougé et al data.

Figure 4 of Rougé et al clearly shows a proposed model of cis-activation. It is also explicitly stated in the Rougé et al. text that while the trans activation model is possible, the trans activation model was not supported by MALS analysis. Furthermore based on the structural analysis, trans activation was unlikely.

To address this issue we suggested in our initial review: "The authors should re-word their manuscript to present a more balanced representation of the Rougé et al. manuscript." We were not alone in this interpretation as Reviewer #2 wrote "Based on their NMR studies the authors imply that the crystal structure [in the Rougé et al. manuscript] is an artifact; however, they provide no alternative structural model for the complex formation."

The authors replied with: "This is correct, we apologize and we have changed our wording to adjust the text." However, we do not agree that the authors have appropriately edited their text. As a result

they misrepresent the results and conclusions from the Rougé et al. manuscript and make claims that are not supported by the published data.

Specifically, in lines 88-92 the authors write: "Intriguingly, this structure suggested that the main interaction between the Ubl45 and CD occurs from Ubl45 on one molecule to the CD in another molecule, suggesting that the activation takes place 'in trans' and that USP7 needs to dimerize for full activity. This deviation from the initial model, along with the small interface between Ubl45 and CD, prompted us to further investigate this interaction and its effect on USP7 activity."

We suggest that the authors re-word lines 88-92 to the following:

"In this structure, it was unclear whether the C-terminal peptide is bound in cis or trans. This ambiguity prompted us to further investigate this interaction and its effect on USP7 activity."

In lines 115-117, Kim et al write: "While this structure convincingly shows the binding of the activating C-terminal peptide, we wondered whether the observed in trans interaction between the body of the Ubl45 domain and the CD could be a consequence of crystal packing."

We suggest that Kim et al re-word lines 115-117 to the following: "this structure convincingly shows the binding of the activating C-terminal peptide, however due to the low resolution of the crystal structures, the disorder of the engineered linker to the C-terminal peptide, and the conformational flexibility of the UBL domains, the authors were unable to determine definitively whether the C terminus of USP7 binds into the activation cleft in cis or in trans."

In lines 133-134, the authors write: "These results suggest that the interaction of CD and Ubl45 could be different from that observed by Rougé et. al. in the crystal structure."

We suggest that the authors re-word lines 133-134 to the following: "These results suggest that a cis interaction between CD and Ubl45 may be more likely."

Indeed, the data from Kim et al also remain ambiguous as in response to reviewer #2 the authors state "...we cannot verify or refute the binding mode observed in the crystal" and in response to reviewer #3 "...the weak Ubl45-CD affinity we favor the interpretation of a weak, dynamic multi-state complex. Nevertheless, alternative scenarios cannot be formally excluded. We have adjusted the text to make our reasoning clearer."

Issue #2: Data discrepancies and lack of direct comparisons between the constructs used in the Kim et al and Rougé et al manuscripts.

We had previously suggested the following in our review:

"Line 193: the hypotheses that Ubl45 Δ C promotes ubiquitin binding could be further supported by NMR data. The affinity of labeled ubiquitin (or p53 peptide-conjugated labeled ubiquitin) could be evaluated with Ubl45 Δ C, and Ubl45 with tail could be added for more clarity."

Kim et al responded with:

"We have addressed this point in a different way, using Michaelis-Menten analysis of the enzyme constructs (new Fig. 3g). When we study activity of CD12345 Δ C on a minimal substrate (UbRho) we observe that it has the same K_M as full length USP7 ($\sim 5 \mu M$) rather than that of CD ($19 \mu M$). Of course the k_{cat} is much lower for this construct (Fig. 3g). We have adjusted the text to explain this point."

However there are several unresolved issues with this approach:

1. The steady state kinetic results for CD12345deltaC contradict data from the Rougé et al. manuscript. Kim et al indicate that CD12345deltaC has a similar K_m to FL USP7, however Rougé et al were unable to determine a K_m for the same construct due to the low activity.
2. Kim et al have not done any studies with the USP7CD+CTP construct, that has a similar K_m and k_{cat} as full-length USP7. Furthermore Kim et al do not reference the USP7CD+CTP construct anywhere in the text. The USP7CD+CTP data partially contradict claims by Kim et al that the activation comes from the UBL45 interaction with the CD.
3. Rougé et al. were unable to obtain accurate kinetic measurements for any construct without the C-terminal peptide because of limited substrate solubility. How were Kim et al. able to overcome this limitation to get K_m value of $18.9 \mu\text{M}$ for USP7CD?

The authors should provide explanations for these discrepancies.

Finally, the authors should provide some explanation for the disconnect between genetic studies indicating that MDM2 rather than p53 is the major physiologic substrate of USP7 (Cummins, J.M., and Vogelstein, B. (2004). *Cell Cycle* 3, 689–692; Cummins et al. *Nature* (2004) 428, 486 <https://doi.org/10.1038/nature02501>), and the mechanistic conclusions based on their model USP7 substrate, ubiquitinated p53.

Response to reviewers:

We thank the reviewers for their efforts and we apologize for any misunderstandings.

- We have implemented all suggestions as detailed below.
- We provide updated figure S1 (accidentally forgotten on the previous revision).
- To improve the visibility of individual residues we have slightly changed the format of figure 2C and E.
- We have made several small changes to improve readability to a general reader as shown in the highlighted version of the manuscript

Reviewers' comments:

Reviewer #1 (Remarks to the Author):

My concerns have been adequately addressed. I see that the current version is improved with better clarity, not just in the kinetic results but also in overall presentation.

Reviewer #2 (Remarks to the Author):

The revised manuscript fully addresses my previous comments. The new set of NMR experiments shows no chemical shift perturbations in UBL45 spectrum upon the CD binding and moderate signal broadening, which authors attribute to a non-specific weak binding between the domains. The UBL45 binding to the CD-Ub, on the other hand, causes more dramatic signal broadening, suggesting a tighter binding in the presence of ubiquitin, in agreement with their other binding experiments. The NMR data showing that the C-terminus of the UBL45 becomes more ordered, however, is not conclusive because the peak broadening upon the CD-Ub binding is quite uniform and affects the C-terminus the same way it affects the N-terminus, unless both, the N- and the C-termini now interact. Apart from a few inconsistencies that can be attributed to the limitations of the methods used, the paper is interesting and warrants publication.

Indeed the peak broadening is quite -but not completely- uniform. The difference in intensity loss is clear when comparing the N-terminal vs. the C-terminal residues, in particular for the terminal residues. We realize that the width of the histogram bars in the original figure were not ideal, so we have made them a more appropriate width.

Reviewer #3 (Remarks to the Author):

Most of the technical comments from reviewers are adequately addressed. Two significant issues remain:

- 1) The authors continue to focus on one model of the crystal structure reported in Rougé et al (Structure 2016, <http://dx.doi.org/10.1016/j.str.2016.05.020>) and indicate that trans-activation is the only proposed mechanism. This limited interpretation is inappropriate as worded in the current form of the Kim et al. manuscript.
- 2) Data discrepancies and lack of direct comparisons between the constructs used in the Kim et al and Rougé et al manuscripts, that should be resolved.

Issue #1: Misrepresentation of the Rougé et al data.

Figure 4 of Rouge et al clearly shows a proposed model of cis-activation. It is also explicitly stated in the Rougé et al. text that while the trans activation model is possible, the trans activation model was not supported by MALS analysis. Furthermore based on the structural analysis, trans activation was unlikely.

To address this issue we suggested in our initial review: “The authors should re-word their manuscript to present a more balanced representation of the Rougé et al. manuscript.” We were not alone in this interpretation as Reviewer #2 wrote “Based on their NMR studies the authors imply that the crystal structure [in the Rougé et al. manuscript] is an artifact; however, they provide no alternative structural model for the complex formation.”

The authors replied with: “This is correct, we apologize and we have changed our wording to adjust the text.” However, we do not agree that the authors have appropriately edited their text. As a result they misrepresent the results and conclusions from the Rougé et al. manuscript and make claims that are not supported by the published data.

Specifically, in lines 88-92 the authors write: “Intriguingly, this structure suggested that the main interaction between the Ubl45 and CD occurs from Ubl45 on one molecule to the CD in another molecule, suggesting that the activation takes place ‘in trans’ and that USP7 needs to dimerize for full activity. This deviation from the initial model, along with the small interface between Ubl45 and CD, prompted us to further investigate this interaction and its effect on USP7 activity.”

We suggest that the authors re-word lines 88-92 to the following: “In this structure, it was unclear whether the C-terminal peptide is bound in cis or trans. This ambiguity prompted us to further investigate this interaction and its effect on USP7 activity.”

We apologize that the introduction still retained the stronger wording, that was unintentional. We are happy to adapt the text as suggested.

In lines 115-117, Kim et al write: “While this structure convincingly shows the binding of the activating C-terminal peptide, we wondered whether the observed in trans interaction between the body of the Ubl45 domain and the CD could be a consequence of crystal packing.”

We suggest that Kim et al re-word lines 115-117 to the following: “this structure convincingly shows the binding of the activating C-terminal peptide, however due to the low resolution of the crystal structures, the disorder of the engineered linker to the C-terminal peptide, and the conformational flexibility of the UBL domains, the authors were unable to determine definitively whether the C terminus of USP7 binds into the activation cleft in cis or in trans.”

We followed the suggestion of the reviewer as follows: “This structure convincingly shows the binding of the activating C-terminal peptide, however the connection to Ubl45 was disordered making it difficult to decide whether the C-terminus of USP7 binds into the activation cleft in cis or in trans”

In lines 133-134, the authors write: “These results suggest that the interaction of CD and Ubl45 could be different from that observed by Rougé et. al. in the crystal structure.”

We suggest that the authors re-word lines 133-134 to the following: "These results suggest that a cis interaction between CD and Ubl45 may be more likely."

We have implemented the suggestion as follows: These results suggest that the interaction of CD and Ubl45 likely takes place in cis.

Indeed, the data from Kim et al also remain ambiguous as in response to reviewer #2 the authors state "...we cannot verify or refute the binding mode observed in the crystal" and in response to reviewer #3 "...the weak Ubl45-CD affinity we favor the interpretation of a weak, dynamic multi-state complex. Nevertheless, alternative scenarios cannot be formally excluded. We have adjusted the text to make our reasoning clearer."

Issue #2: Data discrepancies and lack of direct comparisons between the constructs used in the Kim et al and Rougé et al manuscripts.

We had previously suggested the following in our review:

"Line 193: the hypotheses that Ubl45 Δ C promotes ubiquitin binding could be further supported by NMR data. The affinity of labeled ubiquitin (or p53 peptide-conjugated labeled ubiquitin) could be evaluated with Ubl45 Δ C, and Ubl45 with tail could be added for more clarity."

Kim et al responded with:

"We have addressed this point in a different way, using Michaelis-Menten analysis of the enzyme constructs (new Fig. 3g). When we study activity of CD12345 Δ C on a minimal substrate (Ubrho) we observe that it has the same K_M as full length USP7 ($\sim 5 \mu M$) rather than that of CD ($19 \mu M$). Of course the k_{cat} is much lower for this construct (Fig. 3g). We have adjusted the text to explain this point."

However there are several unresolved issues with this approach:

1. The steady state kinetic results for CD12345 Δ C contradict data from the Rougé et al. manuscript. Kim et al indicate that CD12345 Δ C has a similar K_M to FL USP7, however Rougé et al were unable to determine a K_M for the same construct due to the low activity.

Indeed this construct has low activity (k_{cat} comparable to CD only) and we determine kinetic parameters only when we increased the enzyme concentration to 20 nM. The maximum enzyme concentration Rougé et al used was 10 nM for the more inactive constructs, maybe the 2-fold higher concentration was just enough to get good data (Table S2).

2. Kim et al have not done any studies with the USP7CD+CTP construct, that has a similar K_M and k_{cat} as full-length USP7. Furthermore Kim et al do not reference the USP7CD+CTP construct anywhere in the text. The USP7CD+CTP data partially contradict claims by Kim et al that the activation comes from the UBL45 interaction with the CD.

We are not so sure that there is a contradiction. Rougé et al show that their CD-CTP fusions reach maximally $\sim 60\%$ of full-length activity. Our results agree that the activation comes from the tail, but show that Ubl45 does have an additive role, possibly explaining some of this difference. Our data address what Ubl45 could be doing, although its exact contribution may be difficult to address, since the distance of the tail to the CD with a shorter linker than two Ubl domains may affect the

chance of binding. It is difficult to see how further analysis of the CD-CTP could address this.

However, we do agree that it should be discussed. We have now introduced this : *Intriguingly in this analysis the C-terminal peptide alone, when directly linked can reconstitute much of the activation, but from the structure, it was unclear whether it was bound in cis or trans. This ambiguity prompted us to further investigate the role of the Ubl45 domains in this interaction and their effect on USP7 activity.*

and discussed as follows:

It may also explain why direct linkage of the C-terminal peptide to the CD almost, but not completely recapitulates the full length activity.

3. Rougé et al. were unable to obtain accurate kinetic measurements for any construct without the C-terminal peptide because of limited substrate solubility. How were Kim et al. able to overcome this limitation to get Km value of 18.9 μM for USP7CD?

The authors should provide explanations for these discrepancies.

Rougé et al also report a KM of 19 μM for the catalytic domain (Table S2). As stated in their Methods section and in this rebuttal (see point 1), the usage of a higher enzyme concentration allows for better signal to noise and therefore better determination of these steady state parameters. As explained above, we used two-fold higher concentration of enzyme.

Finally, the authors should provide some explanation for the disconnect between genetic studies indicating that MDM2 rather than p53 is the major physiologic substrate of USP7 (Cummins, J.M., and Vogelstein, B. (2004). Cell Cycle 3, 689–692; Cummins et al. Nature (2004) 428, 486 <https://doi.org/10.1038/nature02501>), and the mechanistic conclusions based on their model USP7 substrate, ubiquitinated p53.

We would like to stress that we have used p53 as a model target protein. In order to do the quantitative analyses described here a homogeneous, well-defined target is essential. Although the recognition sequence for MDM2 is known, the lysine on which the ubiquitin mark is placed is not.

To make the preference of USP7 for MDM2 more explicit in the manuscript we have changed the sentence: *Further definition of the individual interaction sites would give more insights as to how USP7 controls the balance between MDM2 and p53.* into: *Further definition of the different interactions would be needed to explain why USP7 usually prefers MDM2 over p53 (Cummins et. Al., Cummins & Vogelstein).*

REVIEWERS' COMMENTS:

Reviewer #3 (Remarks to the Author):

All of our concerns have been adequately addressed by the authors.